# Optimization and control of actuator networks in variable geometry truss systems using genetic algorithms

Jianzhe Gu[1,2], Ziwen Ye[2], Tucker Rae-Grant [2], Shuhong Wang[1,2], Ding Zhao[3], Josiah Hester[4], Victoria A. Webster-Wood [3] & Lining Yao [1,2] ✉

A robot's morphology is pivotal to its functionality, as biological organisms demonstrate through shape adjustments – octopi squeeze through small apertures, and caterpillars use peristaltic transformations to navigate complex environments. While existing robotic systems struggle to achieve precise volumetric transformations, Variable Geometry Trusses offer rich morphing capabilities by coordinating hundreds of actuating beams. However, control complexity scales exponentially with beam count, limiting implementations to trusses with only a handful of beams or to designs where only a subset of beams are actuable. Previous work introduced the metatruss, a truss robot that simplifies control by grouping actuators into interconnected pneumatic control networks, but relies on manual network design and control sequences. Here, we introduce a multi-objective optimization framework based on a tailored genetic algorithm to automate actuator grouping, contraction ratios, and actuation timing. We develop a highly damped dynamic simulator that balances computational efficiency with physical accuracy and validate our approach with experimental prototypes. Across multiple tasks, we demonstrate that the metatruss achieves complex shape adaptations with minimal control units. Our results reveal an optimal number of control networks, beyond which additional networks yield diminishing performance gains.

Recent studies show that a robot's morphology is pivotal to its functionality[1]. Biological organisms demonstrate this through their ability to adjust body structure and stiffness to accommodate environmental demands – octopi squeeze through small apertures, while caterpillars use peristaltic shape changes to navigate diverse environments. This adaptability becomes indispensable for robots facing tasks such as fitting into tight spaces, conforming to objects, or inducing specific human emotions and cognitive responses. However, creating artificial systems that can adaptively change their shape while maintaining functionality, as seen in these biological examples, remains a significant engineering challenge.

Robotic systems with adaptive morphology demonstrate the ability to change shapes, including continuous volumetric transformations. Bar-joint robots use interconnected bars and joints as linear or rotational actuators, but are often limited to tree topologies[2–4] or single-bar limbs[5], restricting their shape expressiveness and weight-bearing capacity. Multi-material voxel robots[6–9], composed of regular cubic units (voxels) with different material properties, offer diverse shape changes by activating specific voxels. However, they face challenges in scalability and real-world precision due to their solid volume nature and the non-linear interactions between connected voxels. Magnetic self-reconfigurable cubic robots[10,11] allow reassembling of

[1]Morphing Matter Lab, Mechanical Engineering, University of California, Berkeley, CA, USA. [2]Human-Computer Interaction Institute, Carnegie Mellon University, Pittsburgh, PA, USA. [3]Mechanical Engineering, Carnegie Mellon University, Pittsburgh, PA, USA. [4]Interactive Computing and Computer Science, Georgia Institute of Technology, Atlanta, GA, USA. ✉e-mail: liningy@berkeley.edu

voxelated shapes through magnetic connections but lack structural integrity and continuous motion, limiting their practical applications. Other approaches include robots with variable limb lengths[12,13], morphing wheels[14], and 2D origami or soft sheet robots[15,16]. However, these designs often have limited degrees of freedom and control precision or are constrained to specific morphologies, highlighting the need for more versatile and scalable solutions in adaptive robotic systems that can achieve complex, three-dimensional shape changes while maintaining structural integrity and precise control.

Among the various approaches to address these challenges, variable geometry truss (VGT) systems stand out among robotic designs that offer morphological complexity and adaptability. VGTs, composed of beams and joints that form tetrahedral or octahedral truss structures, achieve diverse transformations such as rotation, twisting, linear, and volumetric scaling through actuator beams. This flexibility enables VGTs to perform standard robotic tasks such as locomotion[17,18], manipulation[19,20], and target reaching, as well as specialized activities requiring morphological adaptations[21–24]. Despite their advantages in degrees-of-freedom (DOFs) and versatility, current VGTs face scalability issues due to the complexity of their control systems, which scale exponentially with the number of beams[25]. Therefore, existing VGT with physical implementations are either having a few tetrahedral units[19,26,27] or only a few beams are actuable[21], restricting their achievable motions.

Previously, researchers have introduced an approach to simplify the control of complex truss robots[28], a strategy for grouping actuator air channels into networks, termed C-networks (Control networks). By grouping pneumatic actuators with interconnected joints, each subgroup of actuators in the same C-network can be actuated simultaneously with a single air valve as the controller (Fig. 1a). Although the total number of actuators does not change, the number of controllers decreases. With varying combinations of the actuation states of the C-networks, the metatruss deforms into different morphologies and the number of possible morphologies exponentially scales as the number of C-networks increases (Fig. 1b). Moreover, under a temporal sequence of actuation signals, the truss transforms into a series of morphologies and performs a sequential motion. Additionally, they enabled each actuator to have different preset contraction ratios through a blocker structure (Fig. 1f). This approach aims to simplify the system and control complexities inherent in complex truss robots. Although it introduced a design and simulation tool that allows designers to assign the beam connectivity manually, no optimization or automated design pipeline was introduced. As the truss becomes more intricate and tasks grow in complexity, manually navigating C-network assignment becomes tedious and intractable.

In nature, humans and other animals, despite having hundreds of muscles and billions of muscle cells, execute complex movements without consciously controlling each muscle's contraction. Research indicates that animals may use a control strategy known as synergy[29–32]. This mechanism, also present in humans, reduces neural pathway complexity[31]. With synergy in human motor control, intricate actions such as walking or jumping are executed by periodically coordinating a muscle network, eliminating the need for conscious control of every individual muscle. As such, many muscles operate concurrently when engaging in activities that require collaborative muscle actuation. Although the topic remains under debate, several researchers argue that this coordinated approach achieves an optimal balance between actuator count and control complexity, significantly reducing the computational burden of the brain[33–35]. Inspired by biological muscle synergy, where complex movements are achieved through efficiently coordinated muscle groups rather than individual control, we propose a similar principle for the metatruss. We hypothesize that there exists an optimal number of C-networks for a given metatruss, beyond which additional networks yield a diminishing increase in performance across various multiple tasks.

To validate this hypothesis, we developed a multi-objective optimization pipeline using a tailored hierarchical genetic algorithm (Fig. 1c, d). This approach was chosen because the discrete nature of C-network assignments (Fig. 2) and the topological constraints (Fig. 3) of the network make traditional gradient-based optimization methods unsuitable. Our genetic algorithm (Fig. 4) incorporates custom operators (Fig. 5) that respect both symmetry and connectivity constraints while exploring the design space. The pipeline simultaneously optimizes three parameters: the assignment of actuators to C-networks (determining which actuators work together), the preset contraction levels of individual actuators (defining actuators' motion), and the temporal actuation sequences (controlling when each C-network activates). This multi-level optimization allows us to find designs that balance the competing demands of control simplicity and task performance while maintaining physical feasibility.

In this work, we validate our hypothesis and approach through multiple complementary studies. First, using a complex quadruped metatruss robot (Fig. 1d) tasked with four distinct functions (Fig. 1i, Supplementary Video 1), we demonstrate that a limited number of C-networks can yield competitive performance (Fig. 1j). Our proposed design method achieves a higher ratio of actuatable beams to control units compared to previous VGT systems (Fig. 1k), reducing control complexity while maintaining system capabilities (Fig. 6). To demonstrate the diversity of our approach, we successfully apply it to five different truss topologies with various tasks (Fig. 7). Finally, we build a physical prototype of one metatruss to validate the physical feasibility and compare its trajectory with the simulation, demonstrating the high accuracy of our simulator (Figs. 1l, m, 8).

## Results
### Overview
Here we present our main findings, beginning with the fundamental design of the metatruss system and its optimization framework, followed by experimental validation across multiple studies. This overview first introduces the core mechanisms and design principles, then outlines our key contributions in simulation, optimization, and physical implementation that are detailed in subsequent sections.

Our metatruss, based on the pneumatic shape-changing truss design from PneuMesh[28], is a tetrahedron-based structure composed of pneumatic linear actuators and 3D-printed joints. Each actuator expands to a maximum length under positive pressure $P_+$ and contracts to one of four preset lengths under negative pressure $P_-$, adjustable via a reconfigurable blocker structure (Fig. 1f). The joints have selective inner air channels that connect incident actuators, grouping them into subsets called C-networks (Fig. 1a, b) Actuators within a C-network share air pressure and operate simultaneously, independent of other networks. Each C-network has a binary state: active ($P_+$) or inactive ($P_-$). The metatruss achieves various morphologies through different combinations of C-network states (Fig. 1a, b). Detailed mechanism and fabrication information can be found in Methods - Mechanism and Fabrication Details.

A metatruss can achieve specific shapes or perform sequential motions through activation signals of its C-networks, enabling tasks that require locomotion or shape changes. Whereas previous work[28] demonstrated hand-designed C-network assignments and actuation signals for given tasks, our work automates this process for more complex truss topologies and diverse tasks. Given a metatruss topology, initial joint positions, tasks, number of C-networks, and C-network symmetry, our optimizer finds the optimal C-network assignment, contraction levels, and actuation signals to maximize the metatruss's multi-objective performance across the specified tasks. For a detailed problem definition, refer to Methods - Problem Statement. The specific truss topologies and tasks explored in this paper are described in Methods - Truss Topologies and Tasks.

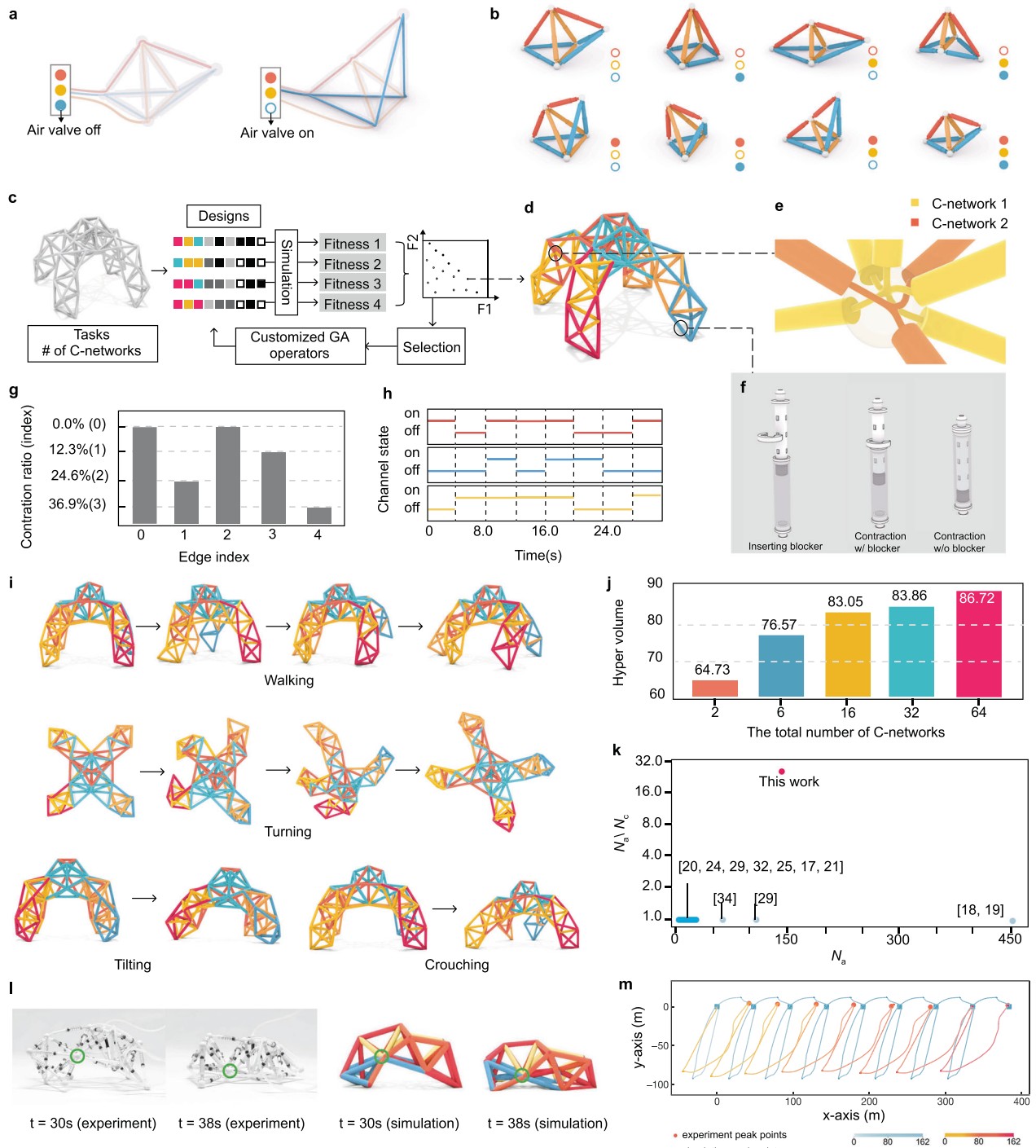

**Fig. 1 | Overview of the metatruss system. a, b** A metatruss with double tetrahedron topology consisting of 9 actuators controlled by three inter-connected air channels (C-networks), where actuators in the same C-network expand simultaneously to yield $2^3$ possible configurations through the combinations of binary C-network actuation states. **c** The multi-objective genetic algorithm with customized operators used for the metatruss design optimization. **d** The C-network assignment, where actuators of the same color belongs to the same C-network. **e** The customized joint structure features inner air channels with selective connectivity, enabling unified control for actuators sharing the same air pressure. **f** Each beam has a discrete contraction level within one of the four percentages

$r \in \{0.0, 0.12, 0.24, 0.36\}$, preset manually with a blocker design. **g** Contraction levels in a metatruss design. **h** Open-loop binary control signals in a metatruss design. **i** Six-channel quadruped metatruss optimized to achieve four distinct target motions through four open-loop controls: walking, turning, tilting, and crouching. **j** Experimental results showing that task performance plateaus as the number of C-network channels increases. **k** Our optimizer achieves higher ratios of actuatable beams ($N_a$) to control units ($N_c$) compared to previous VGT systems[17–22,25–27,37–39,66–69]. **l** The physical prototype of the pillbug metatruss and its simulation, with tracked joint highlighted in green circle. **m** The trajectories of the tracked joint in simulation and experiment.

For clarity, in this paper, metatruss topology refers to the connectivity and structural relationship between joints and beams, analogous to a graph structure. The topology represents the fundamental structure that remains fixed after design, including which beams

connect to which joints and how they're grouped into C-networks. Morphology refers to the physical shape and form that the structure takes when the beams change length. As actuators in the metatruss expand or contract, the positions of joints shift while maintaining their

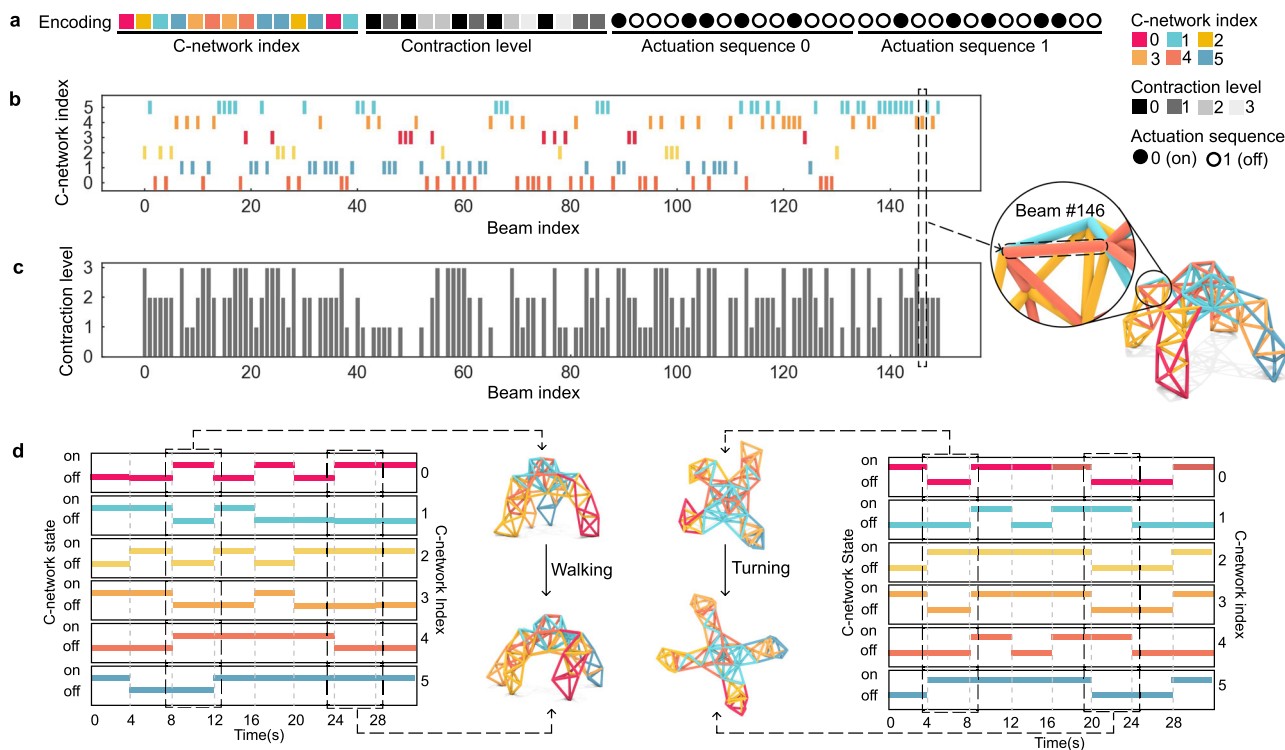

**Fig. 2 | Representation and Elements of a metatruss Design. a** A 1D integer array serving as the design representation. This array encompasses C-network indices, contraction levels, and on/off control signals. **b** In a quadruped robot example, each beam designated a unique C-network index indicated by color. **c** Preset

contraction ratios *r* derived from the product of contraction level and a fixed increment, Δ = 0.12. **d** Task-specific sequences of on/off control signals assigned for actions like walking and turning.

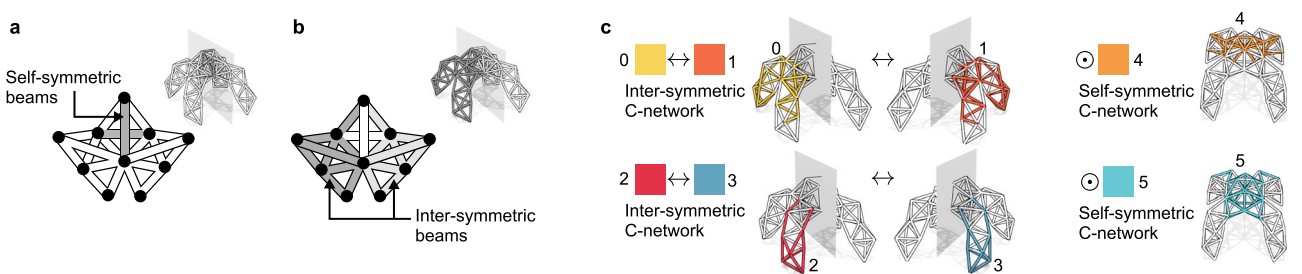

**Fig. 3 | Metatruss symmetry constraints.** A symmetric metatruss consisting of self-symmetric (**a**) and inter-symmetric beams (**b**). **c** Preset C-network configurations designating individual C-network as self-symmetric or specify C-network pairs as inter-symmetric.

topological connections, creating different morphologies that enable the robot to perform various tasks. In this paper, we have a fixed truss topology as input, and we aim to optimize the topologies of C-networks that comprise the entire metatruss, such that the resulting sequential morphological change can achieve the objective behavior.

To optimize the metatruss design, we developed a highly-damped dynamical simulator that balances computational efficiency with physical accuracy. While existing simulators like Finite Element Methods offer high fidelity but are computationally intensive, and pure kinematic approaches are fast but oversimplified, our approach strikes a middle ground necessary for evolutionary optimization. The simulator approximates quasi-static behavior through significant damping while accurately capturing essential physical interactions, including length constraints, gravity, ground collision, and friction. This design choice enables rapid evaluation of thousands of design iterations while maintaining sufficient accuracy for real-world usage, as validated through our physical prototypes. The simulator's performance and accuracy are thoroughly examined in Methods - Simulator, where we demonstrate comparable accuracy to established physics engines

while achieving substantially faster computation times necessary for our genetic optimization pipeline.

Building on our simulator, we developed an optimization framework tailored to the unique challenges of metatruss design. At its core, our approach transforms the complex problem of C-network design into a tractable form by encoding network assignments, contraction levels, and activation signals into a simple yet expressive integer-based representation (see Design Representation). This optimization framework addresses the essential topological constraints of C-networks while enabling efficient evolutionary optimization (see C-network Topology Constraints). To handle multiple competing objectives while maintaining design diversity, we enhanced the NSGA-II algorithm with an elite preservation mechanism (see Multi-objective Computation Pipeline). A key innovation of our framework is its custom genetic operators, which are designed to explore the design space while following the physical and topological constraints (see Tailored Operators).

We validated our framework through three complementary studies. Using a quadruped robot as our primary test case, we

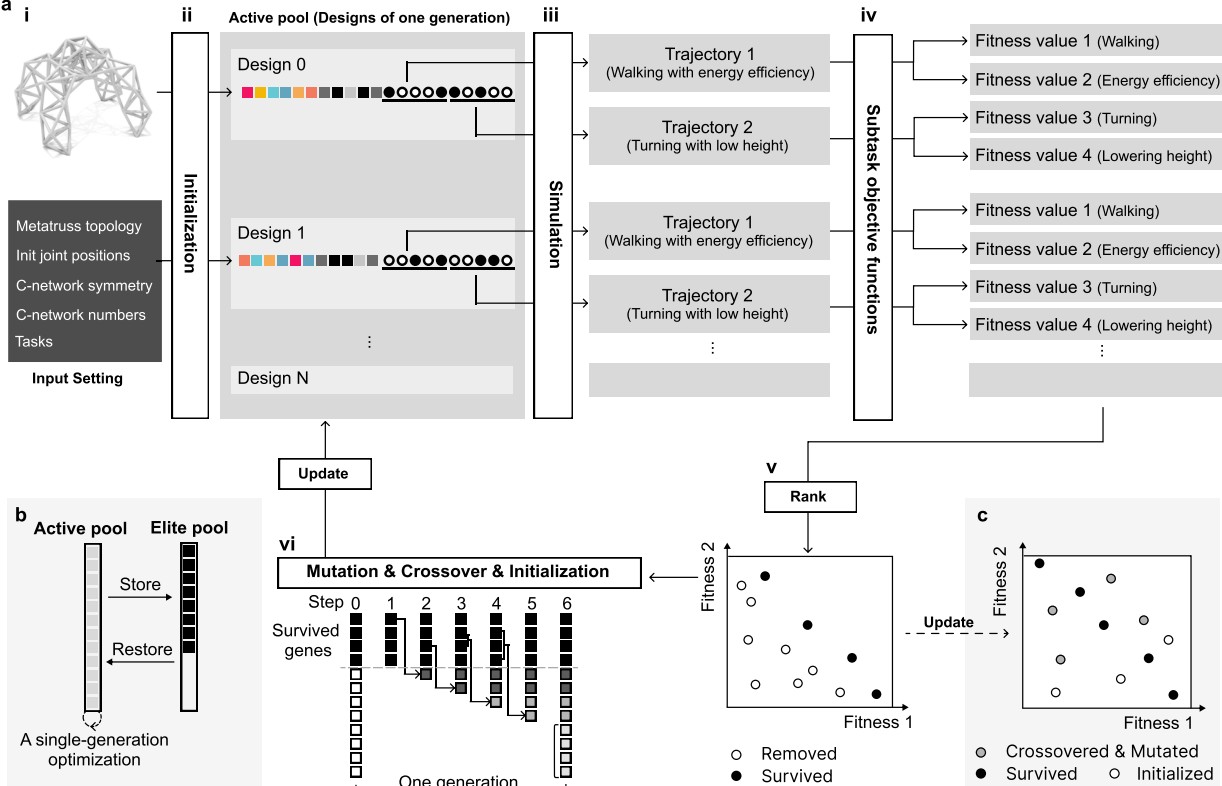

**Fig. 4 | The Optimization Pipeline for metatruss Structures. a** The single-generation optimization involving each training generation to update the active gene pool via NSGA-II-based selection, mutation, crossover, and initialization. (i). The input setting includes the predefined topology, joint positions, symmetry along the *y* = 0 plane, C-network configurations, and objectives. (ii). The initial active gene pool is formulated through a tailored initialization operator. (iii). Different trajectories of the robotic behaviors resulted from simulation. (iv). Within each GA iteration, simulated design trajectories undergo evaluation using specific objective functions. (v). NSGA-II ranks and filters designs, retaining only top performers. (vi). Retained designs generate the next generation via mutation and crossover operators, complemented by additional designs from the initialization operator. **b** The cross-generation optimization to replenish the elite pool from the active pool. For each $N_G$ generation, the remaining designs in the active gene pool are moved to the elite pool, indicating the end of the iteration. Once the elite pool is full, its designs are transferred back to the active gene pool. **c** The performance space of the updated designs, illustrating that operators can generate more superior designs to improve the overall performance of the generation.

demonstrated that metatruss performance reaches diminishing returns beyond a certain number of C-networks, supporting our hypothesis that effective control can be achieved with relatively few networks (see Performance with Varying C-network Channel Numbers). We then showcased the versatility of our approach by optimizing five distinct metatruss designs for diverse tasks ranging from locomotion to shape-morphing (see Diversity in Task and Truss Topology). Finally, we bridged the simulation-reality gap by building and testing a physical prototype, confirming both the practical feasibility of our designs and the fidelity of our simulator (see Physical Validation). Detailed descriptions of the truss designs and their corresponding tasks can be found in Methods - Truss Topologies and Tasks, with numerical results and implementation details in Methods - Numerical Results and Implementation Details.

**Metatruss simulator**

Various approaches have been developed for simulating truss robots, each with their own tradeoffs. Finite element methods (FEM), such as Karamba3D, offer detailed analysis of load distribution and micro-deformations but are computationally intensive[36]. Rigid body simulators like Newton Game Dynamics[21], Open Dynamics Engine[37], Mujoco, and Bullet[2-4] provide a balance of speed and accuracy, making them suitable for interactive use and optimization. Some researchers focus on kinematic analysis, assuming quasi-static motion and fixed contact

points, which allows for simpler inverse kinematics solutions but limits task diversity and dynamic scenarios[17,18,38].

To simulate metatrusses, we employ a highly-damped dynamical simulation model. Although rooted in dynamic simulations, this model utilizes significant damping factors and incremental adjustments to the rest lengths of connecting beams, effectively approximating quasi-static behavior. This approach de-emphasizes the dynamic processes in favor of the final, converged states. The model integrates four types of forces: length-constraint forces, gravity, ground collision, and friction, which are calculated using explicit integration methods. The incorporation of damping ensures that the system approaches a near-equilibrium state at each step, approximating quasi-static behavior while retaining computational efficiency. The model details can be found in Supplementary Note 1: Simulator Details.

As genetic algorithm requires extensive evaluations across multiple generations of designs, and the result needs to be transferred to a physical metatruss robot, our simulator needs to be both efficient and accurate. To evaluate both aspects, we compared our simulator with Mujoco. Using motor actuators and equality constraints in Mujoco, we calculated the root mean square error (RMSE) of joint trajectories between simulators, normalized by total displacement, showing an average difference of 3.60%. Our simulator achieved computation speeds over 340 times faster than Mujoco for 10,000 simulation steps (see Supplementary Note 2: Simulator Comparison).

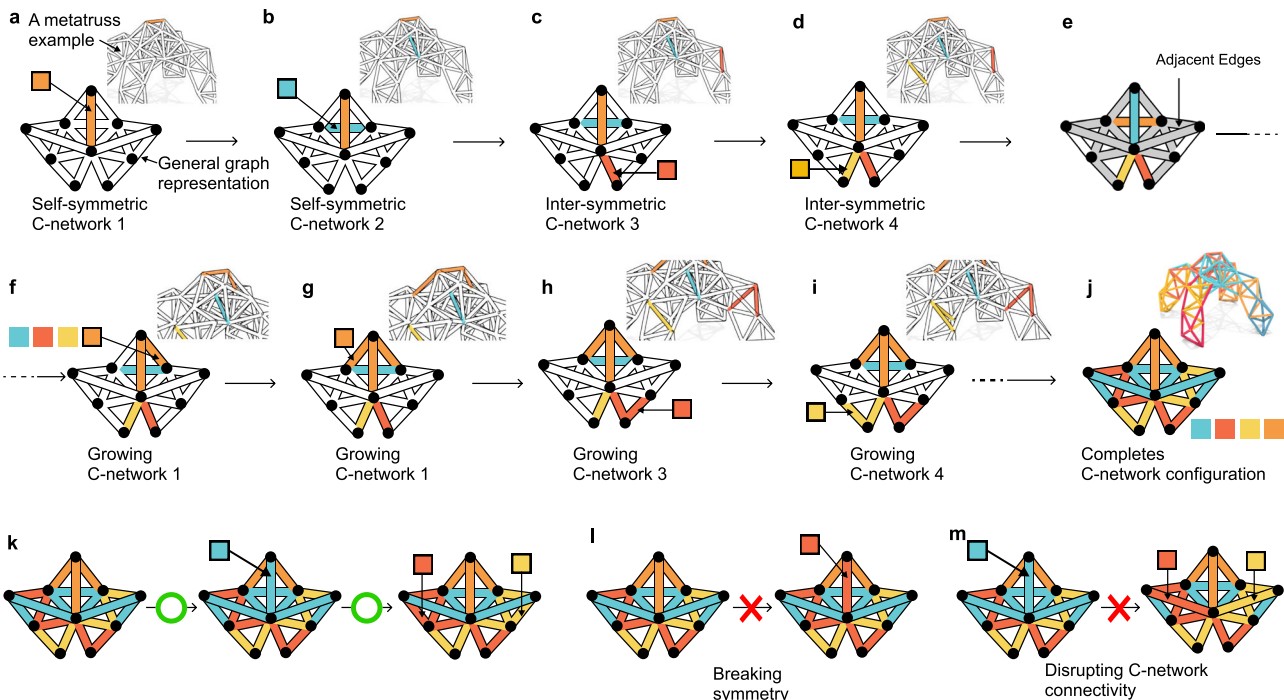

**Fig. 5 | Metatruss Operators including C-network Initialization, Mutation, and Crossover. a–j** A four-C-network initialization process. One beam for each of the four C-networks is randomly selected and assigned a valid C-network index, adhering to C-network and beam symmetry constraints (**a–d**). Beams connected to those already assigned are also assigned with valid C-network indices through iterative selection (**e–j**). **k** The valid mutation steps. The invalid mutations that break symmetry (**l**) or disrupt C-network connectivity (**m**).

To validate our simulator's accuracy, we fabricated a physical prototype of a pillbug-like metatruss. We compared the tracked experimental trajectory with the simulated trajectory over eight action sequence cycles, finding a trajectory difference of 4.38% relative to the total displacement of 86.0 cm (see Physical Validation). This close alignment between experimental and simulation results demonstrates the high accuracy of the sim-to-real transfer of our simulator.

## Optimization framework with tailored genetic algorithm

In the field of VGT, researchers have developed various approaches to optimize the control and motion of truss robots, focusing on the actuation signals for individual beams or joints[17,18,39]. Some studies have explored co-optimization of control and morphology[40–42]. However, these methods typically assume independent control of each actuator and often require continuous contraction ratios, which is not suitable for our metatruss.

Implicit encoding methods have been used to represent element attributes and actions in a continuous latent space. Compositional pattern-producing networks (CPPNs) have been particularly effective for voxel robots[6,43], excelling at generating complex designs with symmetry, repetition, and spatial continuity. These features align well with voxel robots' regular, Euclidean topology. However, truss structures present challenges for CPPNs due to their non-Euclidean topology, where the relationship between neighboring elements is not uniform or continuous in space. Small changes in an actuator's C-network index can dramatically affect metatruss performance or invalidate the structure, and the spatial properties at which CPPNs excel may not be beneficial.

Other approache,s such as the use of transformers[3,4] or L-systems[44] for tree-topology robots, also face limitations when applied to metatruss designs. These methods are well-suited for acyclic, tree-like structures but struggle with the cyclic topology of trusses. Moreover, the number of edges in our metatruss is significantly larger than in typical limbed robots, adding another layer of complexity to the encoding and optimization process.

Graph Neural Networks (GNNs) are naturally suited for cyclic topologies like those in our metatruss system. However, information degradation during message passing has long been a bottleneck[45], and for truss structures, which are supposed to be scalable, the generalizability to more complex structures remains challenging for GNNs.

Given the unique challenges of metatruss optimization—including C-network connectivity constraints, cyclic graph topology, and multi-objective requirements—existing implicit encoding approaches prove inadequate. Instead, we opt for discrete optimization methods, specifically genetic algorithms, which allow direct optimization on explicit encodings. The flexibility of genetic operators enables us to tailor them to our specific constraints. To address the multi-objective nature of our problem, we implement the NSGA-II algorithm (see Supplementary Note 4: NSGA-II Explanation), facilitating simultaneous optimization of metatruss designs across multiple performance criteria.

Our optimization framework takes as inputs the truss topology, initial joint positions, and target objectives. The framework optimizes variables including C-network assignments, contraction levels, and actuation sequences. Each design undergoes evaluation across multiple objective functions, generating performance scores in a multidimensional evaluation space. These scores then feed into the NSGA-II algorithm, which ranks designs based on Pareto dominance and crowding distance to guide selection. The selected designs get transformed through our tailored genetic operators – specifically designed to maintain connectivity and symmetry constraints – to generate the next generation. To prevent premature convergence and maintain design diversity, we implement an elite pool strategy that temporarily preserves high-performing designs while allowing continued exploration of the design space (see Methods - Elite Pool Details).

**Design representation.** To efficiently and concisely describe a metatruss design compatible with the genetic algorithm, we use a one-dimensional integer array as its representation. We represent every parameter, including the C-network assignment, contraction level, and

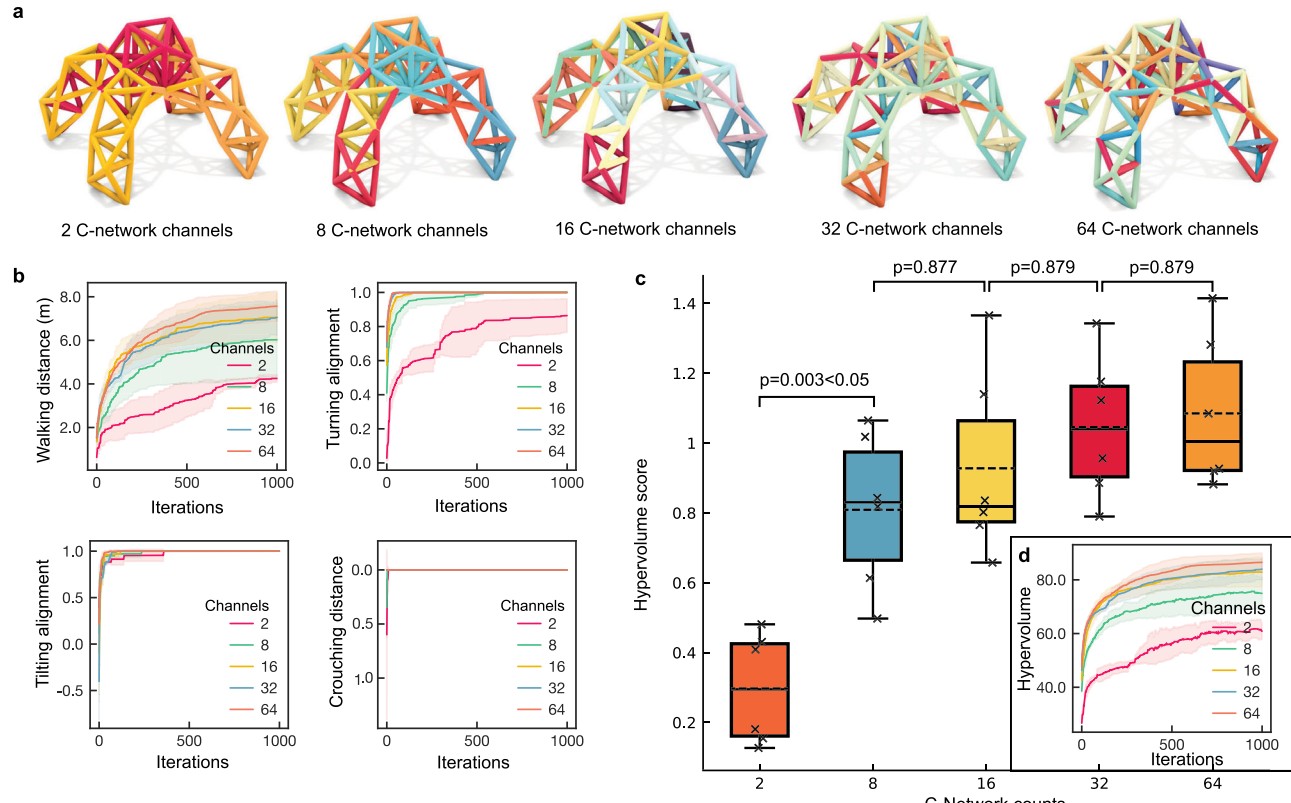

**Fig. 6 | Performance Trade-offs with Varying C-network Channel Numbers.**
**a** C-network assignments for the quadruped robot with 2, 8, 16, 32, or 64 C-network channels at the 1000th iteration. **b** Performance metrics for a quadruped robot for four target objectives, including walking distance, turning alignment, tilting alignment, and crouching distance. **c** Box plots showing hypervolume results for designs with 2, 8, 16, 32, and 64 C-networks ($n = 6$ each). Each box shows the median (solid line) and mean (dahsed line), first and third quartiles (box boundaries), and data range (whiskers). Individual data points are shown as black crosses ( × ).

Horizontal mean lines are displayed within each box. Statistical significance brackets show p-values from Tukey's HSD post-hoc comparisons. ANOVA testing shows significant differences between groups ($p < 0.001$), with post-hoc Tukey's HSD revealing significant improvement from 2 to 8 C-networks ($p = 0.003$) but no significant differences beyond 16 C-networks ($p > 0.877$). **d** Convergence of Pareto front hypervolume across iterations for designs with different numbers of C-networks.

actuation sequences in integers, and concatenate them into a 1D integer vector. Specifically, this array is structured into three segments, each encapsulating specific design parameters of the metatruss (Fig. 2a):

- **C-network Assignment**: The initial segment of the array captures the affiliation of each beam to a specific C-network. Integers within this segment correspond to the indices of C-networks to which each beam is assigned (Fig. 2b).
- **Contraction Level**: The second segment represents the preset contraction levels for the beams. Each integer in this section signifies a preset level, which corresponds to a predefined contraction ratio (Fig. 2c).
- **Actuation Sequences**: The final section captures the dynamic aspects of the metatruss design – the actuation sequences. Each integer here indicates the on/off states for the air valves that govern each C-network at every time step (Fig. 2d). The actuation sequences are flattened into a one-dimensional array and concatenated into the representation.

This encoding represents all the information of a metatruss design as an integer vector that is suitable for the genetic algorithm to optimize. The detailed definition of the representation can be found in Supplementary Note 3: Representation Details.

**C-network topology constraints.** As shown in the design of the actuators and joints(see Methods - Mechanism and Fabrication Details), actuators within the same C-network share the same air

pressure through a continuous air channel network. This implies that a C-network is an undirected connected graph. Any two actuators within the same C-network need to be physically connected, and there exists a path of actuators connecting them, where all the actuators in the path are assigned to the same C-network. We define this as the **connectivity constraint** to ensure the physical validity of the C-networks.

The second is **symmetry constraint**. Symmetry, a concept often observed in nature, has been recognized for its ability to increase the efficiency of robotic movements[46]. By integrating symmetry into the design and control of robots, the parameter space can potentially be substantially reduced, thereby enhancing the search process's efficiency. Here, we define a symmetry constraint during the C-network assignment optimization process. This involves defining symmetry at various levels, including the joint, beam, and truss, and introducing the C-network symmetry configuration (Fig. 3). The details of the definition of the symmetry can be found in Methods - Symmetry Definitions.

The formal definition for the constraints can be found in Methods - Constraint Details.

**Multi-objective computation pipeline.** To optimize toward multiple objectives, a weighted sum of multiple objective function evaluation values is a straightforward solution. However, when different objectives have conflicting requirements, the optimization often ends up at a middle ground, which leads to a solution that is not performing the best at any of the objectives.

One advantage of a genetic algorithm is that instead of optimizing a single design, it optimizes a generation of designs where each design

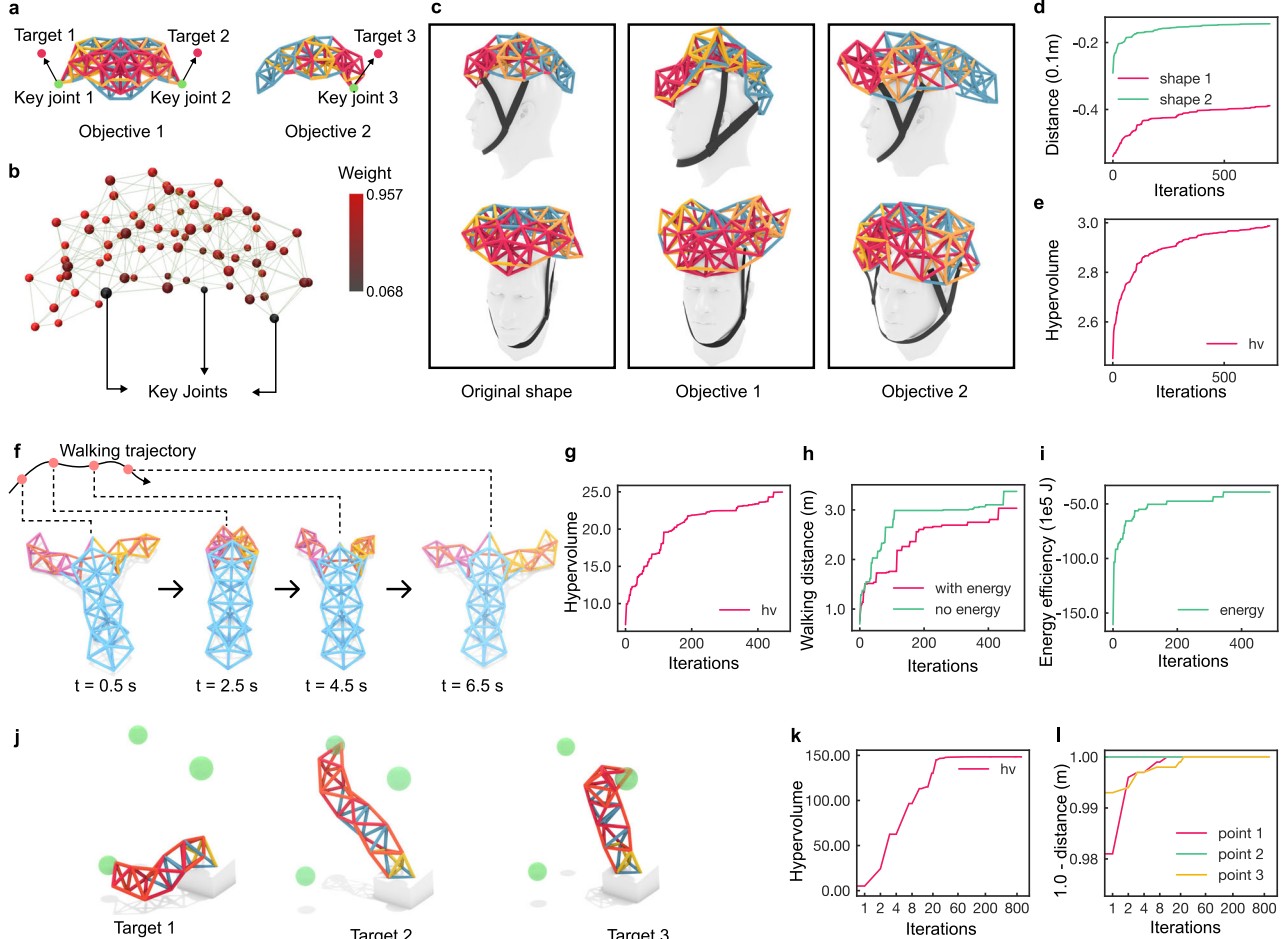

**Fig. 7 | Metatruss Design Variations and Performances. a** Three target positions for a morphing helmet in two objectives, with green indicating the key joints, and red being the corresponding target positions. **b** Joint weights of the morphing helmet are indicated by color shading, with darker shades representing larger weights. **c** The shape of the helmet before and after morphing. **d, e** Shape-shifting helmet training metrics: hypervolume and mean square distance between the key joints and the corresponding target positions. **f** Lobster robot's walking trajectory and morphologies at different times. **g–i** Lobster robot metrics: hypervolume, energy efficiency, and walking distance. **j** Tentacle robot reaching three distinct target positions. **k, l** Tentacle robot metrics: hypervolume and mean squared distance between the tracking joint and three target positions.

has different advantages in some of the tasks. NSGA-II is an algorithm that encourages diversity in the designs through a non-dominated sorting and a crowding distance sorting. Specifically, instead of sorting in one dimension with a weight combination of performances, NSGA-II computes a rank ($R$), which shows to what degree a design is not dominated by other designs, where "dominates" means that a design outperforms another on all objectives. Within the same rank, NSGA-II computes a crowding distance ($CD$) to evaluate how many designs with similar performance exist, and sort them later to encourage design in a sparse performance space to enhance the diversity. The details of NSGA-II can be found in Supplementary Note 4: NSGA-II Explanation.

Each optimization starts with the input of the given topology and initial joint positions, the symmetry and C-network configurations, as well as the objectives (Fig. 4a). A generation of designs is initialized, simulated, and evaluated through multiple objective functions. The resulting objective values are sorted through NSGA-II. The top-performing designs are kept while the rest are discarded. Every few generations, the top-performing designs are moved to an elite pool. Once the elite pool is full, all the elite designs are moved back to the active pool and continue the evolution. The details can be found in Methods - Optimization Process.

In generation-based optimization, a common challenge arises when a subset of designs consistently outperforms others, leading to

their propagation through mutation, crossover, or regeneration operators. As a result, less dominant designs may be prematurely discarded, losing the opportunity to evolve and show their potential. To address this issue, we introduced an elite pool mechanism (Fig. 4b). This approach maintains a separate elite pool in addition to the traditional evolution pool. At regular intervals, the best-performing designs are moved temporarily to the elite pool, creating space for newer designs to evolve in the main pool. After a few iterations, these elite designs are reintroduced to the evolution pool for further optimization, allowing for a more balanced and diverse exploration of the design space and improved Pareto performance. Details of the elite pool mechanism can be found in Methods - Elite Pool Details.

**Tailored operators.** Default genetic algorithm operators do not consider the relationship and constraints between the digits. They randomly generate, change, or exchange the digits within the domain. However, a metatruss representation has symmetry and connectivity constraints, which are not explicitly expressed in the integer vector representation. Therefore, to maintain both the validity of the two constraints and the randomness of the search, we developed custom initialization, mutation, and crossover operators (Fig. 5). For initialization, we need to create a random C-network from an empty metatruss. Instead of randomizing the digits and checking if the

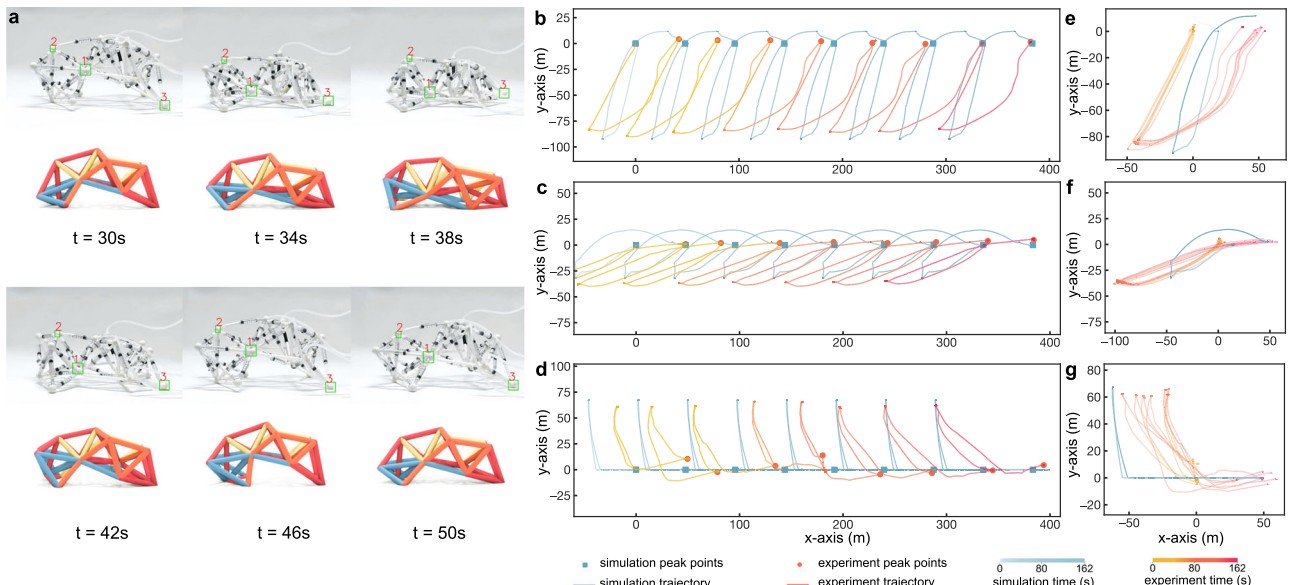

**Fig. 8 | Evaluation of the fabricated pillbug and simulation. a** Comparison of the experimental and simulated pillbug performing walking with lowered body in one action sequence cycle. The three tracked joints are highlighted in green bounding boxes. **b**–**d** The experiment and simulated trajectories of the three joints (e-g: joint 1-3). The color gradient indicates the time of the motion. The joint positions at the beginning and ending of each action sequence cycle are highlighted with peak point markers. **e**–**g** Trajectories of each action sequence cycle from the experiment and simulated trajectories (e-g: joint 1-3). All the cycles are segmented from **b**–**d** at the highlighted points, with starting positions aligned at the zero coordinate.

constraints are valid, we developed a network growth approach. First, it randomly generates new C-network-assigned edges on top of existing C-networks, which assures the connectivity constraint. Meanwhile, the C-network index is filtered on the basis of the symmetry constraints. This ensures the symmetry constraint, connectivity constraints of C-networks, and the randomness of the search. The details can be found in Methods - Operator Details.

**Performance with varying C-network channel numbers**
There exists a trade-off between control complexity and task performance. In an extreme scenario, if the number of C-networks equals the number of beams – implying that each actuator can be controlled independently – the metatruss will possess the maximum DOFs for control, therefore having the potential to achieve optimal performance. However, this is likely unnecessary and generates a tremendously large parameter space (e.g., the metatruss in Fig. 1 would have had 150 air flow control units if each actuator is individually controlled), leading to over-complicated control setup requirements. On the other hand, a robot with too few independently controllable actuators may struggle to perform multiple distinct tasks.

We investigated the relationship between the number of C-networks and robot performance using a quadruped robot model (Fig. 1d). The robot was trained to perform four tasks: walking, turning, tilting, and crouching, with C-network numbers ranging from 2 to 64 (Fig. 6a).

Our results reveal a non-linear relationship between performance and C-network count. For simpler tasks like tilting and crouching, which require only a single-step action, all five robots reach the maximum value (Fig. 6b). In the task of tilting, the 2-C-network robot takes 378 iterations to converge, while the 8-C-network robot and others require 230 or fewer iterations. As tasks become more complex with longer horizons, a performance gap emerges between the 2-C-network, 8-C-network, and 16-C-network robots. Nevertheless, the performance difference diminishes as the number of C-networks increases. For robots with more than 16 C-networks, there is no significant difference in converged performance, with only a 4.8% variation in Pareto front volume among 16, 32, and 64 C-networks (Fig. 6d).

Statistical analysis using one-way ANOVA confirmed significant differences among the five groups ($p < 0.001$, $\eta^2 = 0.138$). Post-hoc comparisons using Tukey's HSD showed that performance improved significantly when increasing from 2 to 8 C-networks ($p = 0.003$), but differences became statistically insignificant beyond 16 C-networks (all $p > 0.877$) (Fig. 6c).

These findings support our hypothesis that an optimized C-network design can achieve competitive performance with a relatively small number of C-networks, balancing task performance with control system complexity. The details of the implementation and analysis can be found in Methods - Numerical Results and Implementation Details.

**Diversity in Task and Truss topology**
To demonstrate the versatility of our metatruss method, we explored a variety of truss topologies and functional objectives beyond simple locomotion tasks. Previous work on Variable Geometry Trusses (VGTs) has primarily focused on single-function designs or limited morphological changes due to control complexity[18,21]. Similarly, other morphing robots have typically been optimized for specific tasks such as locomotion on different terrains or in water[5,47]. Traditional limbed robots, while versatile in movement, are limited in their ability to perform significant shape changes[48].

Our method, in contrast, enables the design of multi-functional, highly adaptable structures while maintaining a simplified control system. It allows for both complex locomotion and volumetric shape morphing. This capability sets our approach apart from both traditional VGTs and limbed robots.

We hypothesized that our approach could optimize trusses for diverse, potentially conflicting objectives within a single design, including both locomotion and shape-approximation tasks. To test this, we developed four distinct examples: a quadruped robot, a shape-shifting helmet, a lobster-inspired walking robot, and a tentacle-like actuator (Figs. 1b, 7).

**The quadruped robot** was optimized for four motion objectives: walking, turning, tilting, and crouching (Fig. 1e, Supplementary Video S1). This demonstration served two purposes: first, to show that

the metatruss could achieve traditional robotic tasks like locomotion and pose changes, and second, to demonstrate that our computational pipeline enabled multiple tasks with a single physical configuration. The robot successfully performed all four motions, with performance improving as the number of C-networks increased up to a threshold value (Fig. 6b).

**The shape-shifting helmet** (Fig. 7a–d) demonstrated our method's capability for precise volumetric shape morphing, successfully transforming between two distinct target shapes while maintaining structural integrity. This capability, which is not typically achievable with traditional limbed robots, could enable robotic functionalities from adapting morphology to meet different environmental constraints and functional requirements to precisely approximating different shapes for esthetic purposes.

**The lobster-inspired robot** (Fig. 7f–i) incorporated energy efficiency alongside locomotion speed, demonstrating improved walking performance and optimization efficiency compared to single-objective optimization. This multi-objective approach extends beyond typical terrain-specific optimizations that focus solely on speed, demonstrating the potential for sustainable locomotion in robots with numerous actuators.

**The tentacle-like actuator** (Fig. 7j–l) achieved high precision in reaching multiple 3D target positions, with error rates below $1e^{-2}$mm for a 173 mm beam length, demonstrating the method's capability for precise shape control. This accuracy suggests promising applications in high-precision manipulation tasks using truss robots.

These examples demonstrate our method's ability to optimize complex, multi-functional truss designs capable of both locomotion and significant shape changes while maintaining simplified control. Our results consistently met or exceeded performance expectations, highlighting our approach's effectiveness in creating versatile, adaptive robotic systems that bridge the gap between traditional limbed robots and highly deformable structures. For detailed information about the topologies and tasks, refer to Methods - Truss Topologies and Tasks.

## Physical validation

To demonstrate the feasibility of our metatruss design and assess the accuracy of our simulator, we constructed and tested a physical prototype called the "pillbug" (Figs. 1l, 8a, Supplementary Video S5). This prototype was designed to perform a walking task with a lowered body, optimized for both locomotion speed and minimized average maximum height. The design was selected from the Pareto front after 800 iterations of training and fabricated using previously established methods[28].

We compared the experimental performance of the pillbug with our simulation predictions by tracking the trajectories of three key joints over eight action sequence cycles (Fig. 8b–d). Our analysis revealed a good overall agreement between the simulated and experimental results, with an average trajectory discrepancy of 3.77 cm and an average static position discrepancy of 1.26 cm. For context, the fully contracted length of each beam in the prototype is 17.3 cm, and the fully extended length is 24.5 cm. The robot achieved a locomotion speed of 2.45 cm/s (approximately 0.048 body length per second), demonstrating effective mobility performance that validates both our control strategy and mechanical design. The total displacement over eight cycles was 86.0 cm.

The prototype demonstrated high consistency in its motion patterns, with an average self-trajectory cycle discrepancy of 0.54 cm. This indicates reliable and repeatable performance despite inherent variances in the physical system, such as friction differences between pneumatic components.

Our results validate the effectiveness of our metatruss design approach and highlight the potential for physical implementation of optimized designs. They also reveal areas for future improvement,

such as accounting for friction variances in the simulation and exploring closed-loop control methods for enhanced accuracy in long-term operation. For more detailed information, please refer to the Methods section, Methods - Physical Prototype and Simulator Accuracy Validation.

## Discussion

In this paper, we present a metatruss design concept with a simplified control system by introducing the C-network mechanism. We develop a tailored multi-objective genetic algorithm to optimize the design of a metatruss under unique design constraints of the system that are discrete and highly relevant to the topology, such that the metatruss can achieve multiple complex motions with a limited complexity of the control system.

We see the potential of our metatruss design method going beyond pneumatically-driven truss robots. Other linear actuators suitable as beams in a truss-robot context could be adapted, including linear actuating beams driven by linear motors, shape memory alloy, and soft actuators such as liquid crystal elastomer[49] or muscle-based biohybrid actuators[50–53]. Another potential direction is to explore metatruss systems without a required physical subnetwork connection. Specifically, a subset of beams can be actuated under the same control signal but does not need to form an interconnected subnetwork. For example, liquid crystal elastomers of different colors can be engineered to respond to remote global lighting with specific wavelengths[54]. In such cases, the constraints from connectivity are alleviated, but the benefits of synergy and reduced control system complexity still stand. This may give us more flexibility on the algorithm side.

Our method shows potential for application in other emerging fields of robotic metamaterials and structures. Recent work has introduced strategies to design and construct classes of robotic metamaterials and 4D printed lattice structures that incorporate complex, multifunctional elements in discrete architectures[55,56]. These approaches create materials capable of outputting multi-DoF motions, sensing capabilities, and programmable thermal and mechanical responses through the manipulation of the properties of local discrete units within 2D or 3D lattices. Our tailored multi-objective genetic algorithm, originally developed for metatruss optimization, could be adapted to optimize these lattice-based structures and potentially automate and speed up the design process, optimizing the arrangement and properties of discrete elements to achieve more complex macro-scale performances or motions while respecting manufacturing and material constraints.

Our metatruss design also shows potential for a fully mechanical implementation of the control system. While our current implementation relies on external control signals, the optimized open-loop control sequences could be encoded directly into mechanical logic circuits as future work. Taking our pillbug robot as an example, its 4-bit binary control sequence with 4 time steps can be implemented using a 2-to-4 multiplexer circuit requiring only 14 logic gates including one clock unit Supplementary Fig. 3. Using pneumatic logic gates based on bistable membranes[57,58], where air pressure differences control the blocking of air tubings, these control circuits could be miniaturized and integrated directly into the metatruss structure. With mechanical logic units potentially scalable to 1cm and metatruss beams expandable to 20 cm, a single metatruss robot could carry its own control circuit board. This approach would significantly simplify the control infrastructure, requiring only a constant air pressure source – either tethered through a single tube or completely untethered with an onboard compressed air tank. This demonstrates how our metatruss design could evolve from externally controlled systems to autonomous, mechanically controlled robots.

Lastly, we can further explore alternative simulators and optimizers using auto-differentiable simulation[42] or density method[59], which

can potentially speed up optimization efficiency through gradient-based methods and provide more design capabilities. For example, the density method may be used to explore an optimal initial topology of the truss structure based on a given three-dimensional mesh, as well as explore the possibility of reconfigurable topology for multi-stage robotic motions or tasks.

## Methods

### Mechanism and fabrication details
The metatruss design builds upon the PneuMesh framework[28], consisting of two key components: pneumatic linear actuators serving as length-changeable beams and specialized joints that connect them. Like other Variable Geometry Truss (VGT) systems, a metatruss achieves shape changes through the coordinated expansion and contraction of these beams. We present the complete mechanism and fabrication details here for comprehensiveness. The metatruss system is fundamentally based on two designs: a discrete-preset-contraction beam system and a selective-air-channel joint network.

**Actuator design**. Each actuator is a syringe-like pneumatic device with an internal air channel and openings at both ends, allowing bidirectional airflow (Fig. Supplementary Fig. 5a, b). Under positive pressure, all actuators expand to their maximum length. Under negative pressure, each actuator can contract to one of several preset lengths.

As shown in Figure Supplementary Fig. 5d, the piston component contains three positioning holes where a C-shaped ring (blocker) can be installed (Fig.Supplementary Fig. 5c). When a negative pressure is applied, the piston retracts until it reaches the blocker. This design enables four discrete contraction ratios: 0% (no blocker), 12%, 24%, and 36%, corresponding to the three blocker positions (Fig.Supplementary Fig. 5d).

This discrete-ratio design serves two purposes. First, it simplifies the parameter space to discrete integers, making it compatible with combinatorial optimization methods. Second, it allows preset morphological variations without increasing the control complexity.

**Joint design and C-network implementation**. The joints, which connect multiple actuators, incorporate an innovative selective-air-channel design (Fig. 1a, b, e). These channels enable specific groups of actuators to share the same air source, ensuring synchronized activation. We refer to these interconnected actuator groups as C-networks (Control networks). Importantly, a single joint can accommodate two independent C-networks without cross-interference, allowing for complex control patterns while maintaining system simplicity.

The joint design process involves several stages of development. First, actuators are assigned to specific C-networks. Then, internal air channels are generated for each C-network passing through the joint. The channel geometry is optimized using Kangaroo, a geometric optimization package, with two primary considerations: maintaining minimum separation between air channels and between channels and outer walls to ensure air-tightness, and minimizing channel curvature to facilitate post-processing removal of support material (Fig. Supplementary Fig. 5e, f). The final joint structure is created using boolean difference operations in Rhino, a parametric design tool (Fig.Supplementary Fig. 5g).

**System constraints and challenges**. While these mechanisms add flexibility to the system, they also introduce several key constraints. All actuators within a C-network must form an interconnected group to satisfy connectivity requirements. The C-networks must maintain prescribed symmetry patterns, and both contraction ratios and control signals are limited to binary/discrete states. These constraints simplify control but create significant challenges for system optimization, particularly as the metatruss scales up in size and complexity.

Navigating this discrete solution space within such a structured framework becomes increasingly challenging with scale.

**Fabrication process**. The fabrication process combines both 3D-printed and off-the-shelf components. We 3D print the pistons, blockers, end caps, and joints using Formlab 3b, while using commercial syringes for actuator shells and rubber tubing for pneumatic connections. The assembly process involves using super glue to construct the actuators, with barb structures 3D-printed directly on ports to secure the friction-fit rubber tubing connections (Fig. Supplementary Fig. 5a, b). For pneumatic actuation, we used an ASLONG AP-370 air compressor (7.73 psi maximum, 0.5–1.5 L/min) for positive pressure and a PYP370 vacuum pump (−7.15 psi, 0.5–2.5 L/min) for negative pressure, controlled via Arduino UNO and electromagnetic valves. Additional components included: Z Rapid iSLA600 3D printer for joints and pistons, silicone tubing (50 durometer, 1.5 mm inner diameter), and polypropylene pipes (6 mm inner diameter) cut using a manual electric grinder.

The completed assemblies are shown in Figure Supplementary Fig. 5, with examples of both the quadruped (Fig. Supplementary Fig. 5h) and pillbug (Fig. Supplementary Fig. 5i, j) configurations demonstrating the versatility of the design.

### Problem statement of metatruss optimizer
**Optimizer**. Given the topology of a metatruss, the initial joint positions, and optimization configurations, which include the N tasks, the lengths of actuation sequences, the number of C-networks, and the symmetry of the C-networks, we want to find an optimal C-network assignment, the contraction level of each actuator, and N open-loop control sequences.

$N$ metatruss motion trajectories will be generated based on $N$ actuation sequences. Each task is a combination of one or more objective functions. And each simulated trajectory will be evaluated by one or more objective functions. $N$ tasks, and $N$ control sequences correspond to $M$ objective functions and $M$ performance scores.

The number of C-networks and the C-network symmetry are given at the beginning of the optimization, where C-network symmetry indicates whether a C-network is self-symmetric of inter-symmetric with respect to another C-network Topology Constraints.

We want to find the optimal C-network assignment, contraction levels, and open-loop control sequences to maximize the performance.

### Symmetry definitions
We define the self-symmetry and inter-symmetry of joints and beams, as well as the symmetry definition of a metatruss (Supplementary Fig.). On top of that, we define the self-symmetry and inter-symmetry of c-networks as the C-network symmetry configuration used for c-network mutation and crossover.

#### Joint Symmetry
*Joint Inter-symmetry*: Two joints are inter-symmetric if they are mirrored to each other against the mirror plane, such that the segment connecting them is perpendicular to the plane and the distance from the two joints to the mirror plane is equal. Inter-symmetry is denoted by $\leftrightarrow$. For example, $v_a \leftrightarrow v_b$ represents for inter-symmetric joints $v_a$ and $v_b$.

*Joint Self-symmetry*: A joint is considered self-symmetric if it is located on the mirror plane. Symbolically, we use $\odot$ to represent self-symmetry. For example, $\odot v_c$ represents that joint $v_c$ is self-symmetric.

#### Beam Symmetry. The symmetry extends to beams as well:
*Beam Self-symmetry*: If both joints of a beam are self-symmetric, the beam itself is deemed self-symmetric (Fig. 3a).

*Beam Inter-symmetry*: A pair of beams is inter-symmetric if the corresponding pairs of joints are inter-symmetric or if one pair is inter-symmetric and the other pair is self-symmetric (Fig. 3b).

We categorize beams into sets $E_s$ for self-symmetric beams and $E_r$ for inter-symmetric beams, satisfying $E_s + E_r = E$.

**Truss Symmetry**. A truss is defined as symmetric if there exists a mirror plane such that every beam is either self-symmetric or inter-symmetric. Although our paper exclusively utilizes symmetric trusses, our algorithm has the flexibility to be applied to asymmetric trusses as well.

**C-network Symmetry**. One of our goals of optimization is to have the C-networks be either self-symmetric or inter-symmetric, which allows symmetric motions. We represent the $i$th C-network by $E_i$, denoting the set of beams sharing the same C-network index $i_C$, where $i_C \in I_C$. *C-network Self-symmetry*: A C-network $E_i$ is considered self-symmetric if $\forall e_m \in E_i$, $\exists e_n \in E_i$ such that $e_m \leftrightarrow e_n$ or $\odot e_m$. *C-network Inter-symmetry*: Conversely, a pair of C-networks $E_i$ and $E_j$ is deemed inter-symmetric if, $\forall e_m \in E_i$, $\exists e_n \in E_j$ such that $e_m \leftrightarrow e_n$. The sum of all C-networks is represented by $\sum_{i=0}^{N_C-1} E_i = E$.

For each optimization task, we predefined the C-network symmetry configuration (Fig. 3c), meaning that we specified the total number of C-networks as well as the count of inter-symmetric and self-symmetric C-networks. Within the C-networks identified as self-symmetric or inter-symmetric (Fig. 3c), we further describe relationships such as $i_{C_0} \leftrightarrow i_{C_1}$, which signifies the C-network with index $i_{C_0}$ is inter-symmetric with the C-network indexed $i_{C_1}$, or $\odot i_{C_2}$, meaning the C-network with index $i_{C_2}$ is self-symmetric.

## Constraint details

**Symmetry constraints**. We define two types of symmetry among joints, beams, and the truss: self-symmetry and inter-symmetry, with respect to a given mirror plane. For joints and beams, inter-symmetry occurs when two joints or beams mirror each other against the plane, while self-symmetry occurs when they mirror themselves (Fig. 3a, b). At the truss level, a truss is defined as symmetric if a mirror plane exists such that every beam is either self-symmetric or inter-symmetric. Although our paper exclusively uses symmetric trusses, our algorithm has the flexibility to be applied to asymmetric trusses as well.

A C-network is also assigned with symmetry in the metatruss. A C-network is considered self-symmetric if it mirrors itself against the middle plane. If two C-networks mirror each other against the plane, they are considered inter-symmetric.

For each optimization task, we predefine the C-network symmetry configuration (Fig. 3c), which means that we specify the total number of C-networks and the count of inter-symmetric and self-symmetric C-networks. Empirically, we set 33% of the C-networks as self-symmetric. For example, in a quadruped robot with six C-networks (Fig. 3c), four C-networks are inter-symmetric, and two C-networks are self-symmetric.

**Connectivity constraint**. In our optimization task, we aim to ensure that all C-networks are connected and that they adhere to either self-symmetry or inter-symmetry in accordance with the predefined C-network symmetry configuration. The connectivity constraint ensures that the C-network forms a network, enabling the transmission of control signals through air pressure. We have three connectivity constraints in the system:

- **Adjacent Beams**: Two beams are termed adjacent if they share a common joint. This connection represents a physical connection between the two beams at that specific joint.
- **Connected Beams**: Building on adjacency, we introduce the concept of connected beams. Two beams are classified as connected if there is a sequence of adjacent beams starting from the first beam and ending at the second. In other words, one can traverse from one beam to the other through this series of adjacent connections.
- **Connected C-network**: Extending the idea to a whole C-network, a C-network is determined to be connected if every pair of beams

within that C-network is connected. This definition guarantees that there's a navigable path between any two beams in the C-network, either directly through adjacency or indirectly via a sequence of adjacent beams.

## Optimization process in one generation with NSGA-II

We introduce our optimization pipeline, a tailored genetic algorithm (GA) that is illustrated in Fig. 4 by using the previously introduced quadruped robot (Fig. 1d, i) as an example. The design of a metatruss is situated in a discrete combinatorial space. The truss structure is inherently a network composed of multiple subnetworks – C-networks – that have shared nodes, but their edges are exclusive. The design of subnetworks involves C-network indices, contraction levels, and binary control sequences, which are all discrete in nature. While there are methods that treat discrete variables in a continuous manner, often used in topology optimization, such as a density-based approach[60], they do not fit our problem due to the connectivity constraint we impose: all beams within the same C-network need to belong to the same subnetwork in a truss network to ensure they are interconnected. This constraint is not easily expressed in a continuous and differentiable form, which conventional optimization algorithms would require.

Given this constraint, we turn to the GA[61], a method adept at handling discrete and combinatorial search spaces. To ensure our specific constraints are respected, we base on the standard GA and customize its operators, enabling it to efficiently explore the design space while following our C-network connection constraints and the discrete nature of parameters.

GA is first used to initialize a generation of designs $\{D_0, D_1, \ldots, D_{N_G}\}$, where $N_G$ is the population size of one generation (Fig. 4a i, ii). Each design is simulated following $N_A$ action sequences, which are evaluated by the corresponding subtask objective functions and yield $N_s$ fitness values (Fig. 4a iii-v).

We use Non-dominated Sorting Genetic Algorithm II (NSGA-II)[62] for selecting designs, keeping some designs, and removing others. To fill up the gene pool again, we use mutation and crossover on the kept designs, and add the newly generated ones through mutation, crossover, and initialization into the kept pool (Fig. 4a vi). This renewal of the gene pool increases the chance of the algorithm reaching higher-performing designs over time, potentially moving them to a new Pareto Front.

## Elite pool strategy for optimization across generations

At this stage, we introduce two distinct gene pools: an active gene pool with a capacity of $N_a$ designs, and an elite gene pool with a capacity of $N_e$ designs (Fig. 4b). Each generation, the designs in the active gene pool are assessed and sorted using the NSGA-II algorithm, based on Pareto dominance and crowding distance. A fixed percentage $\rho$ of top-performing designs, referred to as elite designs, are preserved, while the rest are discarded. The active gene pool is then updated with new designs generated through crossover, mutation, and regeneration operators.

Every $N_g$ generations, instead of simply preserving the elite designs within the active gene pool, these elite designs are temporarily moved to the elite gene pool. This allows the remaining non-elite designs to continue evolving, providing them with the opportunity to further optimize and potentially exceed the current elite designs.

Once the elite gene pool reaches its capacity, the designs it contains are moved back into the active gene pool. This cyclical process encourages competition between both elite and non-elite designs. The elite gene pool thus serves two key purposes: initially, it protects high-performing designs, preventing premature convergence, while promoting diversity and exploration. Later, as elite designs are reintroduced into the active pool, it drives further exploitation of advantageous designs for high-quality solutions.

By alternating between the preservation of elite designs and their reintegration, this mechanism helps balance exploration and exploitation, ultimately leading to better Pareto performance (Fig. 4c).

In our paper, we use $N_e$ equal to $N_a$, and $\rho = 20\%$. We move elites to the elite pool every $N_g = 5$ generations. Therefore, the elite pool becomes full and is returned to the active pool every $N_g/\rho = 25$ generations. The detailed number of $N_a$ for each metatruss can be found in Numerical Results and Implementation Details.

## Operators

With the constraints of symmetry and connectivity in place, traditional GA operators fall short. The standard processes, such as initialization that creates a new gene mutation that randomly selects and alters digits within a single gene, and crossover that involves swapping digits between two genes, do not align with the unique dependencies introduced by the symmetry and connectivity constraints in the metatruss. Simply applying the standard GA operators would violate symmetry and connectivity within the C-networks and beams. Therefore, we must employ customized operators that are tailored to these constraints. In the following sections, we will introduce three such operators that have been designed to function within the constraints of our optimization problem.

**Initialization of C-network Indices**. Standard initialization methods in GA are inadequate to navigate the unique challenges posed by our system's C-network configurations and symmetry constraints. Therefore, we have designed a specialized initialization operator that respects both symmetry and connectivity constraints.

The algorithm starts with an unassigned truss structure graph and progressively employs the C-network configurations. The first step involves earmarking a single beam for each C-network based on its symmetry property (Fig. 5a–d). Specifically, beams selected for self-symmetric C-networks are self-symmetric; for inter-symmetric C-networks, a beam and its inter-symmetric counterpart are chosen simultaneously.

Once this anchor layer of beams is assigned, the algorithm moves to the iterative phase (Fig. 5e–j). Here, it selects unassigned beams that are adjacent to already-assigned beams. The C-network assignment for these beams adheres to two key criteria: i) they must be adjacent to a beam that shares the same C-network, and ii) their symmetry properties must align with the chosen C-network. This ensures that both connectivity and symmetry constraints are satisfied.

This iterative process continues until no unassigned beams remain, thereby initializing designs that are feasible and optimized for the subsequent stages of the genetic algorithm. The detailed algorithm for the initialization and assignments of C-network indices is explained in Algorithm: Initialization Operator.

**Mutation**. The mutation process introduces randomness into the C-network connections. It may involve changing the C-network assignment of a beam while maintaining the connectedness of beams in the same C-network. The aim here is to allow the exploration of the solution space beyond the initial population and to prevent the algorithm from getting stuck at local optima (see Algorithm: Mutation Operator).

Given the unique constraints of our problem, a specialized mutation operator (Fig. 5k) is necessary for effective optimization. Governed by a pre-defined probability $p_m$, the mutation process aims to explore the design space while ensuring C-network connectivity. During the mutation process (Fig. 5k), given a design $D_m$ from survived designs, a random beam $e \in D_m$ has a $p_m$ chance to be selected for mutation. The beam's C-network may be altered, subject to the following conditions: i) The new C-network index $i_m$ must be one of the adjacent C-network indices, and ii) If $e$ is self-symmetric, the new C-network $i_m$ must also be self-symmetric. Otherwise, there is no inter-

symmetric beam for the inter-symmetric C-network. If $e$ has an inter-symmetric beam $e'$, the C-network index of $e'$ will be altered accordingly.

A fail-safe checking mechanism will be applied each time a mutation is applied. If the mutation results in the disconnection of a C-network, the operation will be reverted (Fig. 5l, m). The iterative process will continue until a random number $r$ exceeds $p_m$. This operator ensures that the mutation is both random and constrained, facilitating the traversal of the design space without violating the system's structural or functional integrity.

**Constrained crossover operator for design synthesis**. The crossover operation (see Algorithm: Crossover Operator) leverages two randomly selected surviving designs as parent designs, aiming to create offspring with features from both. The process is analogous to the mutation operator but involves the exchange of beams between two designs instead of altering beams within a single design.

Briefly, a beam $e_i^0$ is randomly picked from the first parent design, and its corresponding beam $e_i^1$ in the second parent is identified. If applicable, the C-network indices of the two beams are then swapped, along with their inter-symmetric counterparts. A fail-safe checking ensures that the swapping adheres to the symmetry and connectivity constraints. The operation iterates until a successful crossover is achieved, thereby synthesizing new designs while preserving the requisite constraints.

## Truss topologies and tasks

**Quadruped robot**. The quadruped robot was designed with 150 actuatable beams arranged in a symmetric configuration. The robot's structure consists of four legs connected to a central body, maintaining bilateral symmetry along its longitudinal axis. This symmetry was reflected in the C-network assignments, with 30% of the C-networks designated as self-symmetric and the remainder as inter-symmetric pairs.

The quadruped was optimized for four distinct tasks: 1. Walking: Forward locomotion along the x-axis. 2. Turning: 90-degree rotation around the z-axis. 3. Tilting: Changing the orientation of the robot's top surface. 4. Crouching: Lowering the overall height of the robot.

These tasks were chosen to demonstrate the robot's ability to perform diverse movements using a single optimized configuration. The performance of the quadruped across these tasks with varying numbers of C-networks is detailed in the Performance with Varying C-network Channel Numbers section.

**Shape-shifting helmet**. We designed a shape-shifting helmet with two functional objectives. The two objectives represent specific target shapes the robot is trained to achieve. For each objective, the goal is to let the assigned key joints approximate the corresponding target positions (Fig. 7a), while maintaining the rest of the non-key joints at their original locations as much as possible. When computing, we assign each joint a weighting factor. The value of the weight is based on each joint's proximity to the closest key joint along the beams. The weight of each joint diminishes as it moves further from the key joints, and the key joints themselves carry the maximum weight (Fig. 7b).

Figure 7c shows the resultant shape transformations. As demonstrated, the helmet effectively approximates each target shape (Supplementary Video S2). The training performance corresponding to each objective is depicted in Fig. 7d,e. The plots show that the morphing helmet's ability to approximate target shapes improved steadily over the iterations.

**Lobster robot trained for energy efficiency**. We use a lobster-inspired walking robot to study how energy efficiency can be integrated into the functional objective (Fig. 7f, Supplementary Video S3). The robot that walks with energy efficiency has two subtasks. One

subtask is to achieve high locomotion speed by evaluating the displacement of the centroid after the action sequence is completed. The second subtask is to minimize the energy consumption of the robot, which is calculated by accumulating the axial force and displacement of all joints along time steps (see Supplementary Note 5: Objective Functions section for details).

Figure 7g–i shows the lobster performance with iterations. We observed that, although all the metrics are improving, using multiple objectives can enhance the search efficiency and quality. For example, when searching for a solution for both the locomotion task and energy efficiency task, the search is faster and converges at a better result than searching for locomotion only (Fig. 7h).

**Tentacles.** We also demonstrated how the metatruss method could be utilized to design a tentacle actuator that approaches multiple target locations in a volumetric space with its end joint (Fig. 7j), with its training performance reported (Fig. 7k, l).

The joint at the most distal position of the tentacle is designated as the key joint. The tentacle is assigned three objectives, each including one subtask of reaching the target point. The three subtasks are to reach three different target points, each situated at the centers of the green dots (Fig. 7j, Supplementary Video S4).

This task demonstrates the metatruss's effectiveness in tasks requiring precision. By the 800th iteration, the key joint is able to achieve proximity to each target position with a distance of less than $1e^{-2}$mm, with each beam extending to a maximum length of 173mm. The tentacle shown in Fig. 7j is one of the designs selected from the Pareto front, exhibiting the closest distances to the three respective target positions, with deviations of 6.01mm, 2.06mm, and 3.37mm. This design is picked based on the smallest standard deviation of the three fitness values. It indicates the tentacle's ability to closely approach all three targets within a single design despite the inherent conflict of reaching three points.

**Pillbug robot.** While the previous examples demonstrate the diversity of achievable morphologies and tasks, the Pillbug robot was designed specifically to showcase the physical feasibility of our metatruss design and to assess the accuracy of our simulation. The Pillbug was optimized for two tasks similar to those of the quadruped robot: walking forward and lowering its body.

The robot's structure resembles a pillbug but with four legs, featuring two smaller forelegs and two larger back legs. It consists of 50 actuatable beams, arranged to allow for both locomotion and body posture changes. We assigned an objective with two subtasks: firstly, to achieve a high locomotion speed, and secondly, to minimize the average maximum height of the metatruss robot. This combination of subtasks was chosen to create a low-profile, efficient walking motion.

After optimization, we selected one design from the Pareto front for physical fabrication. The fabrication process followed the approach described in our previous study[28]. The physical prototype was actuated to complete its action sequence across eight cycles, with the trajectory tracked across three of its joints for comparison with the simulated results.

Detailed information about the Pillbug robot's design, fabrication, and the comparison between experimental and simulated results can be found in the Physical Prototype and Simulator Accuracy Validation section.

### Numerical results and implementation details

**Implementation details.** In our study, we use the notation $\leftrightarrow$ to indicate inter-symmetric channels and $\odot$ to denote self-symmetric channels. We implemented various metatruss designs with different C-network configurations. The quadruped design with 8 channels features a symmetric topology, with channels configured as $0 \leftrightarrow 1$, $2 \leftrightarrow 3$, $4 \leftrightarrow 5$, $\odot 6$, $\odot 7$, comprising 6 inter-symmetric and 2 self-symmetric

channels. A simpler quadruped design with 2 channels maintains symmetry with both channels being self-symmetric ($\odot 0$, $\odot 1$).

We also explored more complex quadruped designs: a 16-channel version with 4 self-symmetric and 12 inter-symmetric channels, a 32-channel version with 8 self-symmetric and 24 inter-symmetric channels, and a 64-channel version with 16 self-symmetric and 48 inter-symmetric channels. The helmet design exhibits symmetry with three self-symmetric channels ($\odot 0$, $\odot 1$, $\odot 2$), while the lobster design is symmetric with two inter-symmetric and two self-symmetric channels ($0 \leftrightarrow 1$, $\odot 2$, $\odot 3$). Notably, the tentacle design is asymmetric, consisting of four asymmetric channels (0, 1, 2, 3).

In our optimization process, we set the active pool size equal to the elite pool size, with an elite percentage of 20% in every generation. For each iteration, we retained 50% designs, generating 35% new designs through mutation and 10% through crossover operations, while introducing 5% new random initializations. The pool size is scaled according to the C-network numbers: 128 for the 8-channel quadruped, 48 for the helmet, 64 for both the lobster and tentacle, 32 for the 2-channel quadruped, 256 for the 16-channel quadruped, 512 for the 32-channel quadruped, and 1024 for the 64-channel quadruped.

We conducted our computations using Google Cloud Computing with 224 cores, achieving an average optimization time of 8 hours and 2 minutes for the 8-channel quadruped over 1000 iterations.

**Performance analysis with varying C-network numbers.** To investigate the relationship between the number of C-networks and robot performance, we conducted a series of experiments using a quadruped robot model. The robot was trained to perform four distinct tasks: walking (maximizing forward distance), turning (90-degree rotation), tilting (changing top orientation), and crouching (lowering height). We tested configurations with 2, 8, 16, 32, and 64 C-networks, always setting 30% as self-symmetric and the rest as inter-symmetric. This 30% ratio was chosen empirically, and future work could explore the optimal ratio for different robot configurations and tasks.

The genetic algorithm was run for 1000 iterations for each configuration. We used the active pool size of 128, with 64 designs retained after each iteration. New designs were generated through mutation (45 designs) and crossover (13 designs), with 6 new random initializations per iteration. The elite pool capacity was also set to 128.

Performance was evaluated using the hypervolume of the Pareto front at the 1000th iteration. The hypervolume metric was chosen as it provides a scalar measure of the quality of a Pareto front in multi-objective optimization, capturing both the spread and the proximity to the ideal point. It was calculated using the PyGMO library's hypervolume function, with a reference point set to the worst observed values for each objective plus a small offset. We conducted a one-way ANOVA to compare performance between C-network numbers, with the significance level set at 0.05. Tukey's Honest Significant Difference (HSD) was used for post-hoc pairwise comparisons.

The ANOVA assumptions were verified: data independence was ensured by separate tests, homogeneity of variance was confirmed by Levene's test (F = 0.085, p = 0.986), and normal distribution was verified by the Shapiro-Wilk test (F = 0.962, p = 0.343).

ANOVA results showed significant differences among the five groups ($F_{(4, 10)}$ = 12.840, p < 0.001, $\eta^2$ = 0.138). Tukey's HSD revealed significant performance improvement when increasing from 2 to 8 C-networks (p = 0.003), but no statistically significant differences among configurations with 16 or more C-networks (p > 0.877). Complete pairwise comparison results are provided in Supplementary Table S1.

### Physical prototype and simulator accuracy validation

We conducted a comprehensive evaluation of our physical prototype, the "pillbug", to validate our metatruss design and assess the accuracy of our simulator.

**Experimental setup and data collection**. We manufactured and assembled the pillbug prototype following the process detailed in Mechanism and Fabrication Details. For the experimental setup, we placed the pillbug on a flat table covered with white photography background paper (HUAMEI brand). The video recording setup consisted of a camera positioned 85 $cm$ horizontally from the table's center and elevated 25 cm above the table surface, with the lens oriented horizontally. We recorded at a 60 Hz frame rate. For lighting, we installed two camera lighting panels approximately 70 cm from the subject at 45° and −45° angles, positioned 40 $cm$ above the table surface. The trajectory of the pillbug is through the center of the table.sssssssss.

The pillbug prototype was actuated to complete its action sequence across eight cycles, with the entire process recorded on video (Fig. 8a, Supplementary Video 5). We tracked the trajectories of three key joints throughout the experiment, collecting 536 tracked points for each action sequence cycle (Fig. 8b–d).

To facilitate comparison with our simulation, we overlaid the simulated trajectory on the experimental data. We highlighted static positions at the beginning and end of each cycle to identify key points of motion and rest. The trajectory was then segmented into individual cycles at these static positions for detailed analysis.

**Data analysis and metrics**. To quantify the similarity between simulated and experimental results, we calculated several metrics:

*Average Trajectory discrepancy* ($d_t$): The mean point-wise distance between corresponding points on the experimental and simulated trajectories, averaged across the entire trajectory.

*Average Trajectory Cycle discrepancy* ($d_c$): The mean point-wise distance between each experimental trajectory cycle and the corresponding simulated cycle, averaged across all cycles and trajectories.

*Average Static Position discrepancy* ($d_s$): The mean point-wise distance between each pair of static positions across all trajectories.

*Average Self Trajectory Cycle discrepancy* ($d_f$): To assess the internal consistency of the prototype, we calculated the mean trajectory cycle for each joint and measured the distance between each experimental cycle and this mean.

**Results and Conclusion**. Our analysis yielded the following results: $d_t$ = 3.77 cm (averaged across all three trajectories), $d_c$ = 2.68 cm, $d_s$ = 1.26 cm, $d_f$ = 0.54 cm. For reference, the fully contracted length of each beam in the prototype is 17.3 cm, and the fully extended length is 24.5 cm. The average total body length during movement is 51.4 cm, with an average locomotion speed of 2.45 cm/s (approximately 0.048 body length per second). The total displacement is 86.0 cm.

The larger values of $d_t$ and $d_c$ compared to $d_s$ likely result from inherent friction differences in the pneumatic plastic syringes used as beams, causing unsynchronized actuation times under uniform air pressure. This factor was not accounted for in our simulation. The smaller $d_s$ indicates good alignment at static positions, which may be attributed to our use of a highly-damped dynamic model and relatively low actuation pressure (+0.8 psi and −0.8 psi) in the simulation.

The fact that $d_t$ is higher than $d_c$ suggests accumulated errors across cycles, a common phenomenon in open-loop control systems. The low $d_f$ value indicates high consistency in the metatruss motions despite inherent friction variances.

These results validate our metatruss design approach while highlighting areas for future improvement, such as standardizing linear actuators for uniform friction or integrating friction variance into the simulator. Additionally, the findings suggest that incorporating sensors for closed-loop control could enhance long-term accuracy, particularly given the ample interior space in the metatruss design.

## Usage of large language model in writing

In the preparation of this manuscript, we utilized ChatGPT with GPT-4 specifically for grammar checking purposes. The prompts we employed were structured as follows: "[the text] Please check the grammar of this writing as a submission for a scientific journal, please don't change or distort the meaning or create new information". All edited text was reviewed and finalized by the human authors.

## Data availability

All training performance histories and tracked joint trajectories used in this study are available, along with code for loading and plotting, at: https://github.com/morphing-matter-lab/MetaTruss-data[63]. Requests related to data can be made to L.Y. Source data are provided with this paper.

## Code availability

Codes for genetic algorithm, tracking, and data processing are available at: https://github.com/morphing-matter-lab/MetaTruss[64,65].

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

## Acknowledgements

We thank Sam Kriegman, Tate Johnson, Guanyun Wang, and Yuyu Lin for their insightful suggestions. Research was sponsored by the Army Research Office and was accomplished under Cooperative Agreement Number W911NF-23-2-0138 (LY, VWW, JH). The views and conclusions contained in this document are those of the authors and should not be interpreted as representing the official policies, either expressed or implied, of the Army Research Office or the U.S. Government. The U.S. Government is authorized to reproduce and distribute reprints for Government purposes, notwithstanding any copyright notation herein. The authors also acknowledge support from the National Science Foundation Career Grant IIS2427455 (LY).

## Author contributions

J.G. and L.Y. conceived the initial concept. L.Y., J.H. and V.W. supervised the project. J.G., L.Y. wrote the manuscript. J.G., Z.Y. performed and implemented the tailored genetic algorithm, conceived and trained five demonstrations, and conducted experiments on the effect of C-network count on performance. J.G., T.R.-G. and S.W. performed the fabrication of the pillbug. J.G. performed pillbug locomotion tracking. J.G. conducted ANOVA data analysis and simulation-to-real comparison. D.Z. provided a server for computation. D.Z., J.H., V.W.-W. and L.Y. provided scientific and experimental advice. All authors commented on the manuscript.

## Competing interests

A provisional patent has been filed with the U.S. Patent Application No. 18/845,842, and all authors are listed as co-inventors. Otherwise, the authors declare no competing interests.
