## [Transparent Peer Review file · Nature Communications]

Optimization and Control of Actuator Networks in Variable Geometry Truss Systems Using Genetic Algorithms

Corresponding Author: Professor Lining Yao

Version 0:

Reviewer comments:

Reviewer #1

(Remarks to the Author)

This paper presents a evolutionary optimisation framework to design the actuation/controller layout within variable geometry metatrusses.

The method is an interesting and logical extension of the group's previous work in this space, which shows promising results.

My major criticism is that I found the paper is very challenging to understand and decipher. Critical features of the work were not clearly outlined and left to the reader to infer for example: the novelty and specific contribution of this work; its relevance to the field and relationship to other works in both VGTs and evolutionary robotics; and the application and utility of this method.

The method itself I found difficult to understand, what exactly was being optimized and how was somewhat mirky. For example the fact that the base geometry was prespecified and not part of the optimization problem, and that the lab has previously published work on the design and manufacture of metatrusses was not evident to me until I was deep into the paper. Throughout the paper key concepts are also dropped with little context or stated as fact without adequate support. In my opinion, the writing and presentation needs to be substantially edited to give greater clarity, to bring this work upto a publishable standard.

It would also be valuable to compare the method with more compact design encodings to highlight its value within evolutionary robotics.

Detailed comments below:

1. The introductory section of this paper is very sparse, it gives a high level description of the morphing robotics and then jumps straight into talking specific features if VGTs. This section should be greatly expanded to give more background on relevant work in evolutionary and morphing robotics
2. The flow of the paper is hard to follow, with methods, computational experiments and physical implementation all blended together. It would be better to reorganise to have the methods together (including all the example problems and their description), then numerical results (optimiser efficacy, convergence, support for synergy hypothesis), then experimental results (with a description of the physical implementation of the device).
3. The numerical results lack key information about the optimisation problem and performance. E.g what is the generation size and number of retained elites, how is the diversity metric implemented? What was the computation time?
4. I found the concept and implementation of eliteness somewhat confusing, could you clarify how the 'elite' candidates are stored and released.
5. The implementation of the EA problem is obviously useful, but it's a sort of brute-force approach. It would be interesting to compare the direct encoding of the networks you've used to a shape grammar or CPPN, which could more compactly encode the design domains and directly embed constraints (rather than randomly searching until a valid solution is found at each iteration). This might allow for larger metatrusses to be generated, or for the morphology (i.e truss configuration) and control to be cooptimized.
6. Another interesting feature is the idea of muscle synergy you highlighted: your system clusters together groups of 'muscles' to reduce the total number, but each has an independent control signal. Can you extend this to show how the muscles can coordinate by responding a central signal? i.e have a single repeating signal that coordinates all the muscles in the way that a regular heart beat triggers a series of muscles to expand and contract in sequence.

7. The figures were often used as evidence for features which were not shown or hard to understand – they need to be modified to give a clearer explanation of the optimisation framework. E.g.:

- a. “These joints have inner air channels (Fig. 1a)” – 1a illustrates the idea of shape change using VGTs, but no air channels are shown
- b. “metatruss system also incorporates the design of a discrete-preset contraction beam (Fig. 1c).” – Fig 1c doesn't really show anything in my opinion, illustrating this preset configuration needs more than a bar chart with discrete steps. For both this and 6a it would be beneficial to show the mechanical implementation of the device rather than a very loose representation (and better yet move that to an experimental implementation section)
- c. The writing seems to conflate the actuator layout with truss design, it should be clear that the truss itself (what I would call the morphology) is not optimised, just the actuator layout and cycle (the controller, in my opinion). E.g Fig 4a shows topology as part of the optimisation, presumably this means C-network topology, it needs to be clarified that this is not VGT topology
- d. Figure 4(vii) gives the impression that fitness is reevaluated after the mutation/crossover step, but (if I'm not mistaken) this occurs at 4(iv)
- e. “sorting them using NSGA-II based on their rank and crowding distance.” – crowding distance is not defined here but is key to the idea of eliteness

(Remarks on code availability)

Reviewer #2

(Remarks to the Author)

I congratulate the authors to this delightful paper and highly interesting approach and novel methodology.

Overall paper:

Very good paper excellent approach.

Language:

High level.

Abstract:

- Precise abstract with all important points

Biological Inspiration and Our Hypothesis:

- Page 3 figure 1g please increase the font size of the references to match the size of the font in 1f. In addition please standardize the axis titles as there is a mix of words starting with capital letters (c,f, etc.) and in (d) only the first word starts with a capital letter
- Page 4 line 79 there is an “an” missing before “empirical”.

Mechanism and model:

- Page 4 line 88 change: “validation for” to “validation of”
- Please add an explanation of “NSGA-II”.

Tailored Design Configurations and Constraints:

- Nothing to add

Computation Pipeline:

- Nothing to add

Performance with Varying C-network Channel Numbers:

- Figure 6 switch the lettering of subfigures c) and d)

Metatruss Designs and Experimental Validation:

- If possible I would like to see a physical demonstrator for the shape shifting helmet.
- Page 16 line 388 please change Figure 7g-h to 7g,h as used above for 7d,e (page 15 line 378).
- Page 17 line 415 there is a space missing Fig.8a
- The pillbug demonstrator is very impressive. Although it was not the focus of the experiment but could you please measure the walking speed of the demonstrator and add it to the text?

Conclusion:

- Please briefly elaborate on the chosen examples and the results achieved with them in terms of expectance, performance and usefulness as system capabilities demonstrations. Could there be other demonstration cases?

Methods:

- Figure 9 fonts and font sizes are all different please standardize.
- Page 23 line numbers are missing for the first paragraph and the figure reference is wrong should be Fig. 9 not fig.10 as Figure 10 describes a different system.

Figures:

- Very good. Please see above hints.

(Remarks on code availability)

Reviewer #3

(Remarks to the Author)

The authors develop a novel framework and optimization approach to design multifunctional truss systems and validate in on an experimental pneumatically driven pill bug. The approach seems like a good advance for this field and seems like a good fit for the journal.

The only feedback I have is for the authors to consider other emerging and related systems where this work seems to be applicable, such as 3D printed robotic metamaterials (<https://www.science.org/doi/10.1126/science.abn0090>) and 4D printed lattice-based robots (<https://onlinelibrary.wiley.com/doi/abs/10.1002/adma.202307858>), and discuss how these tools could be applied for such systems.

(Remarks on code availability)

For some reason I could not view or run the code. Instructions on this would be most helpful.

Version 1:

Reviewer comments:

Reviewer #1

(Remarks to the Author)

Thank you for the comprehensive feedback, I believe the authors have done a commendable job of addressing my concerns from the first version.

The work was always technically very sound, but in my opinion needed to clarify the methodology and contributions. I believe this has now largely been achieved.

My only outstanding comments are:

- 1) I think it would be valuable to define the meaning of 'morphology' and 'topology' in this paper early on as this would immediately dispel any confusion about which parts of the network/structure are included in the optimization
- 2) The phrase metatruss seems to be used in several contexts. i.e in the abstract metatruss is referred to as "an optimization framework", but in the remainder of the paper it seems to refer to the VCG topology and control. Could you clarify its intended meaning? Also the term should be defined in the main text
- 3) Several of the section links are broken in the overview section (lines 149-165)

(Remarks on code availability)

Reviewer #2

(Remarks to the Author)

Dear authors,

Thank you for responding to all comments and changing the manuscript accordingly.

The only issue that is left is that in the overview the references seem to be off there is ?? were a number should be.

Otherwise all comments were met and the manuscript was improved above and beyond what was requested. Especially the on-body control system is an intriguing idea.

Nothing to add.

(Remarks on code availability)

Reviewer #3

(Remarks to the Author)

I appreciate the time and care the authors put in their responses to reviewers. I have only a couple of minor points of feedback.

1. I could not see the authors' need for their rebuttal figure 1 (i.e., I did not find anywhere in their text where they reference rebuttal figure 1.

2. I noticed that, in their responses to the second reviewer, the authors added locomotion speed. I appreciate this, and think that it might be even better to report the speed in terms of body length per unit time so they can have a direct comparison to biology.

(Remarks on code availability)

I can confirm that I was able to download and view the code. It appears that there is an extensive README file with appropriate instructions. I did not get a chance to download and run this myself, but I did not see any glaring issues from a cursory review.

Response to Reviewers

This document contains our point-by-point responses to all reviewer comments and concerns. For clarity of presentation, we use the following color coding scheme:

Reviewer comments are shown in black text. Our responses to these comments are highlighted in blue. The changes made to the manuscript and supplementary materials are indicated as follows: light green text denotes minor revisions and unchanged text, while green text highlights substantial modifications and key changes.

We greatly appreciate the thoughtful feedback provided by the reviewers. Each comment has been carefully addressed, and the corresponding changes have been implemented in the revised manuscript and supplementary materials.

Referee #1 (Remarks to the Author):

Comment #1

This paper presents an evolutionary optimisation framework to design the actuation/controller layout within variable geometry metal trusses.

The method is an interesting and logical extension of the group's previous work in this space, which shows promising results.

Our Response: We thank the referee for the positive comments!

Comment #2

My major criticism is that I found the paper is very challenging to understand and decipher. Critical features of the work were not clearly outlined and left to the reader to infer for example: the novelty and specific contribution of this work; its relevance to the field and relationship to other works in both VGTs and evolutionary robotics; and the application and utility of this

method.

The method itself I found difficult to understand, what exactly was being optimized and how was somewhat murky. For example the fact that the base geometry was prespecified and not part of the optimization problem, and that the lab has previously published work on the design and manufacture of metatrusses was not evident to me until I was deep into the paper. Throughout the paper key concepts are also dropped with little context or stated as fact without adequate support. In my opinion, the writing and presentation needs to be substantially edited to give greater clarity, to bring this work up to a publishable standard.

Our Response: Thanks for your detailed feedback. We have separated each point the reviewer mentioned and addressed them one-by-one as follows:

In my opinion, the writing and presentation needs to be substantially edited to give greater clarity, to bring this work up to a publishable standard.

Our Response: We made a thorough restructuring and rewriting of the paper to improve the writing clarity. Please check Comment #5 for more details.

that the lab has previously published work on the design and manufacture of metatrusses was not evident to me until I was deep into the paper.

Our Response:

Thank you for the feedback! We now clarified the contribution of the previous research PneuMesh, the challenges and how our work addressed the challenges. The previous work PneuMesh introduced the basic mechanism of syringe-based actuators and 3D-printed joints with interconnected air channels, along with preset discrete contraction blocker structures for actuator control. It provided an online editor for manual design, simulation and visualization, where users could assign C-networks, set contraction levels, and edit actuation signals while viewing the resulting motions through a JavaScript-based simulator.

However, this approach had several limitations: the simulator lacked the efficiency needed for optimization and could not be easily parallelized, the manual design process became intractable for complex structures due to inherent C-network topology constraints, and implementations were limited to relatively simple structures like a lobster robot with only 50 beams, of which 30 were passive.

In this paper, we presented an optimization framework that automates the design process of

the actuator grouping, contraction ratio and control signals, and proved the effectiveness and generalizability through a few experiments and demonstrations.

Our Modifications: We clarified the relationship between our work and previous research through additions throughout the manuscript. In Introduction paragraph 4, we now introduce the PneuMesh framework and identify the key limitations that motivated our current work. Please see the revised paper or Comment #4 for detailed changes in Introduction.

The *Results-Overview* section begins with a comprehensive description of the physical system and mechanisms established in PneuMesh.

Our metatruss, based on the pneumatic shape-changing truss design from PneuMesh [1], is a tetrahedron-based structure composed of pneumatic linear actuators and 3D-printed joints. Each actuator expands to a maximum length under positive pressure P_+ and contracts to one of four preset lengths under negative pressure P_- (Fig. 9c,d), adjustable via a reconfigurable blocker structure (Fig. 1f). The joints have selective inner air channels that connect incident actuators, grouping them into subsets called C-networks (Fig. 1a,b) Actuators within a C-network share air pressure and operate simultaneously, independent of other networks. Each C-network has a binary state: active (P_+) or inactive (P_-). The metatruss achieves various morphologies through different combinations of C-network states (Fig. 1a,b). Detailed mechanism and fabrication information can be found in the *Methods-Mechanism and Fabrication Details*.

The *Methods-Mechanism and Fabrication Details* section provides complete technical specifications of the fabrication process. A “Metatruss Mechanism” figure is added in the section to show linear actuator design, the discrete preset contraction ratio mechanism, and the joint design (The new figure is quoted in Comment #11).

Mechanism and Fabrication Details

The metatruss design builds upon the PneuMesh framework [1], consisting of two key components: pneumatic linear actuators serving as length-changeable beams and specialized joints that connect them. Like other Variable Geometry Truss (VGT) systems, a metatruss achieves shape changes through the coordinated expansion and contraction of these beams. We present the complete mechanism and fabrication details here for comprehensiveness. The metatruss system is fundamentally based on two designs: a **discrete-preset-contraction beam system** and a **selective-air-channel joint network**.

Actuator Design

Each actuator is a syringe-like pneumatic device with an internal air channel and openings at both ends, allowing bidirectional airflow (Fig. 9a,b). Under positive pressure, all actuators expand to their maximum length. Under negative pressure, each actuator can contract to one of several preset lengths.

As shown in Figure 9d, the piston component contains three positioning holes where a C-shaped ring (blocker) can be installed (Fig. 9c). When a negative pressure is applied, the piston retracts until it reaches the blocker. This design enables four discrete contraction ratios: 0% (no blocker), 12%, 24%, and 36%, corresponding to the three blocker positions (Fig. 9d).

This discrete-ratio design serves two purposes. First, it simplifies the parameter space to discrete integers, making it compatible with combinatorial optimization methods. Second, it allows preset morphological variations without increasing the control complexity.

Joint Design and C-Network Implementation

The joints, which connect multiple actuators, incorporate an innovative selective-air-channel design (Fig. 1a,b). These channels enable specific groups of actuators to share the same air source, ensuring synchronized activation. We refer to these interconnected actuator groups as C-networks (Control networks). Importantly, a single joint can accommodate two independent C-networks without cross-interference, allowing for complex control patterns while maintaining system simplicity.

The joint design process involves several stages of development. First, actuators are assigned to specific C-networks. Then, internal air channels are generated for each C-network passing through the joint. The channel geometry is optimized using Kangaroo, a geometric optimization package, with two primary considerations: maintaining minimum separation between air channels and between channels and outer walls to ensure air-tightness, and minimizing channel curvature to facilitate post-processing removal of support material (Fig. 9e, f). The final joint structure is created using boolean difference operations in Rhino, a parametric design tool (Fig. 9g).

what exactly was being optimized and how was somewhat mirky. For example the fact that the base geometry was prespecified and not part of the optimization problem

Our Response: To clarify the optimization framework: the truss topology (base geometry) is provided as input, along with the desired objective tasks and channel configurations. The optimizer then determines three key elements: the grouping of actuators into control networks (C-network assignment), the contraction ratio for each actuator, and the open-loop control sequences.

Our Modification: We revised Figure 4 to separate predefined parameters from optimization variables by adding an input parameter block at the top left of the diagram. We added a new section Methods-Problem Statement and referred to it in Results-Overview to specify that the truss topology is provided as input while C-network assignment of each actuator serves as an optimization parameter. Both the Problem Statement and revised Figure 4 show the distinction between fixed inputs and optimization variables. For additional details, please refer to Comment #12.

the novelty and specific contribution of this work, Throughout the paper key concepts are also dropped with little context or stated as fact without adequate support

Our Response: This work makes several key contributions: first, we introduce a computational pipeline that fully automates the design optimization process. This enables handling significantly more complex structures, as demonstrated by a quadruped robot with 150 actuated beams performing four distinct motions. This pipeline includes a tailored genetic algorithm that can optimize designs under multiple competing objectives while respecting physical constraints, and a simulator that is efficient for optimization and accurate for physical experiments. Second, we validate our approach through multiple demonstrations across different topologies and tasks. Finally, we empirically show that performance gains diminish beyond a certain number of control networks, supporting our hypothesis about optimal control complexity.

Our Modification: We reorganized the manuscript structure to present the contribution of this work more clearly. Introduction paragraph 6 describes our contributions, input parameters, and optimization variables, followed by experimental results and validation in the subsequent paragraph.

To validate this hypothesis, we developed a multi-objective optimization pipeline using a tailored hierarchical genetic algorithm (Fig. 1c,d). This approach was chosen because the discrete nature of C-network assignments and the topological constraints of the network make traditional gradient-based optimization methods unsuitable. Our genetic algorithm incorporates custom operators that respect both symmetry and connectivity constraints while exploring the design space. The pipeline simultaneously optimizes three parameters: the assignment of actuators to C-networks

(determining which actuators work together), the preset contraction levels of individual actuators (defining actuators' motion), and the temporal actuation sequences (controlling when each C-network activates). This multi-level optimization allows us to find designs that balance the competing demands of control simplicity and task performance while maintaining physical feasibility.

We validate our hypothesis and approach through multiple complementary studies. First, using a complex quadruped metatruss robot (Fig. 1d) tasked with four distinct functions (Fig. 1i, Supplementary Video 1), we demonstrate that a limited number of C-networks can yield competitive performance (Fig. 1j). Our proposed design method achieves a higher ratio of actuatable beams to control units compared to previous VGT systems (Fig. 1k), reducing control complexity while maintaining system capabilities. To demonstrate the diversity of our approach, we successfully applied it to five different truss topologies with various tasks. Finally, we built a physical prototype of one metatruss to validate the physical feasibility and compare its trajectory with the simulation, demonstrating the high accuracy of our simulator (Fig. 1l,m).

For a detailed discussion of the Introduction modifications, please refer to Comment #4.

The *Results* section begins with a newly added Overview that introduces each component sequentially: a C++ based simulator for efficient simulation, an optimization framework based on genetic algorithms with integer vector representation of designs, and formal definitions of symmetry and connectivity constraints.

We reorganized the rest of *Results* as follows: 1). We introduced the computation framework that incorporates NSGA-II implementation with elite mechanisms and tailored operators. 2). We validated our approach through three experimental studies: an analysis of C-network numbers and performance relationships, demonstrations across multiple truss topologies and tasks, and physical validation using a pillbug prototype.

We moved implementation details, numerical analysis, and fabrication specifications into *Methods* and *Supplementary Information*, which are referred in the relevant places in Results.

For the unchanged sections, we refined the writing based on reviewer feedback to improve clarity and readability.

its relevance to the field and relationship to other works in both VGTs and evolutionary robotics;

Our Response: We addressed the paper’s relationship to VGT and evolutionary robotics fields through revisions to the Introduction section. We expanded our literature review to comprehensively discuss morphology-changeable robots, including VGT implementations, evolutionary robots, and other approaches. For specific modifications and details, please refer to Comment #4.

Comment #3

It would also be valuable to compare the method with more compact design recordings to highlight its value within evolutionary robotics.

Our response: Thank you for the insightful suggestion. Our understanding is that compact design recordings refers to more compact representation of the design parameters, which are usually also continuous and implicit. Please correct us if there is any misunderstanding or confusion. We agree that a compact, continuous or implicit representation can potentially enable the utilization of continuous optimization and creates a compact latent space for more efficient search than direct encoding like integer arrays. The two methods we thought that have the highest potential are CPPN and VAE. We added a new discussion about the potential usage and limitations of CPPN, implemented a VAE and reported the new results. Please refer to Comment #8 for details.

Comment #4

The introductory section of this paper is very sparse, it gives a high level description of the morphing robotics and then jumps straight into talking specific features of VGTs. This section should be greatly expanded to give more background on relevant work in evolutionary and morphing robotics

Our Response: Thank you for this valuable suggestion. We agree that the original introduction lacked sufficient background on evolutionary and morphing robotics, making the transition to VGTs abrupt.

Our modifications:

After the first paragraph, which remains as a general introduction to morphing robotics, we have added a new second paragraph. We examine robots with morphological changes, such as evolutionary robots and morphing robots, and explain the challenges in the limited degree of freedom (DoF) and complexity in control and simulation.

The review covers limbed robots and graph-topology robots using bar-joint components, which are often limited to tree topologies or single-bar limbs. Multi-material voxel-based robots optimized by evolutionary algorithms offer shape changes through specific voxel activation but face challenges in scalability and real-world precision due to their solid volume nature and non-linear interactions between connected voxels. Magnetic self-reconfigurable cubic robots enable shape reassembly through magnetic connections but lack structural integrity and continuous motion. Additional approaches include robots with variable limb lengths, morphing wheels, and 2D origami or soft sheet robots, though these designs often have limited degrees of freedom and control precision, or are constrained to specific morphologies.

Robotic systems with adaptive morphology demonstrate the ability to change shapes, including continuous volumetric transformations. Bar-joint robots use interconnected bars and joints as linear or rotational actuators, but are often limited to tree topologies [2–4] or single-bar limbs [5], restricting their shape expressiveness and weight-bearing capacity. Multi-material voxel robots [6–9], composed of regular cubic units (voxels) with different material properties, offer diverse shape changes by activating specific voxels. However, they face challenges in scalability and real-world precision due to their solid volume nature and the non-linear interactions between connected voxels. Magnetic self-reconfigurable cubic robots [10, 11] allow reassembling of voxelated shapes through magnetic connections but lack structural integrity and continuous motion, limiting their practical applications. Other approaches include robots with variable limb lengths [12, 13], morphing wheels [14], and 2D origami or soft sheet robots [15, 16]. However, these designs often have limited degrees of freedom and control precision or are constrained to specific morphologies, highlighting the need for more versatile and scalable solutions in adaptive robotic systems that can achieve complex, three-dimensional shape changes while maintaining structural integrity and precise control.

In the next paragraph of the introduction, we discuss how VGTs offer unique advantages including volumetric morphing capability, large degrees of freedom, ease of simulation and control, and structural stability, which address the limitations of existing morphing robots. However, we also point out that current VGTs face challenges with cubically scaled control complexity, which has limited physical implementations to simple topologies. We merged the previous two paragraphs about VGT background into this single focused paragraph to simplify the presentation while maintaining all key content.

Among the various approaches to address these challenges, variable geometry truss (VGT) systems stand out among robotic designs that offer morphological complexity and adaptability. VGTs, composed of beams and joints that form tetrahedral or oc-

tahedral truss structures, achieve diverse transformations such as rotation, twisting, linear, and volumetric scaling through actuator beams. This flexibility enables VGTs to perform standard robotic tasks such as locomotion [17, 18], manipulation [19, 20], and target reaching, as well as specialized activities requiring morphological adaptations [21–24]. Despite their advantages in degrees-of-freedom (DOFs) and versatility, current VGTs face scalability issues due to the complexity of their control systems, which scale exponentially with the number of beams [25]. Therefore, existing VGT with physical implementations are either having a few tetrahedral units [19, 26, 27] or only a few beams are actuatable [21], restricting their achievable motions.

In the fourth paragraph, we introduce our previous work PneuMesh and its metatruss mechanism. Inspired by biological synergy mechanisms, we developed an actuator-grouping VGT approach using C-networks, which demonstrated the potential to perform complex tasks with a reduced number of control units. While this approach helped balance control complexity and performance, we point out that the lack of an automated design tool limited the scaling of both truss morphology complexity and task complexity. The discrete nature of truss topology and its inherent constraints made manual design tools insufficient as complexity increased.

Previously, researchers have introduced a novel approach to simplifying the control of complex truss robots [1], a strategy for grouping actuator air channels into networks, termed C-networks. By grouping pneumatic actuators with interconnected joints, each subgroup of actuators in the same C-network can be actuated simultaneously with a single air valve as the controller (Fig. 1a). Although the total number of actuators does not change, the number of controllers decreased. With varying combinations of the actuation states of the C-networks, the metatruss deforms into different morphologies and the number of possible morphologies exponentially scales as the number of C-networks increases (Fig. 1b). Moreover, under a temporal sequence of actuation signals, the truss transforms into a series of morphologies and performs a sequential motion. Additionally, they enabled each actuator to have different preset contraction ratios through a blocker structure (Fig. 1f). This approach aims to simplify the system and control complexities inherent in complex truss robots. Although it introduced a design and simulation tool that allows designers to assign the beam connectivity manually, no optimization or automated design pipeline was introduced. As the truss becomes more intricate and tasks grow in complexity, manually navigating C-network assignment becomes tedious and intractable.

The content of the fifth, sixth and seventh paragraphs remain largely unchanged. In the fifth

paragraph, we present our hypothesis: given tasks and meta-truss topology, the relationship between the number of C-networks and meta-truss performance is non-linear, with performance gains becoming marginal beyond a certain number of C-networks. We then introduce our main contribution in the sixth paragraph: a discrete optimizer based on a tailored genetic algorithm that optimizes C-network assignment, preset actuator contraction ratios, and open loop control signals of a given truss topology for multiple tasks.

In nature, humans and other animals, despite having hundreds of muscles and billions of muscle cells, execute complex movements without consciously controlling each muscle's contraction. Research indicates that animals may use a control strategy known as synergy [28–31]. This mechanism, also present in humans, reduces neural pathway complexity [30]. With synergy in human motor control, intricate actions such as walking or jumping are executed by periodically coordinating a muscle network, eliminating the need for conscious control of every individual muscle. As such, many muscles operate concurrently when engaging in activities that require collaborative muscle actuation. Although the topic remains under debate, several researchers argue that this coordinated approach achieves an optimal balance between actuator count and control complexity, significantly reducing the computational burden of the brain [32–34]. Inspired by biological muscle synergy, where complex movements are achieved through efficiently coordinated muscle groups rather than individual control, we propose a similar principle for the meta-truss. We hypothesize that there exists an optimal number of C-networks for a given metatruss, beyond which additional networks yield a diminishing increase in performance across various multiple tasks.

To validate this hypothesis, we developed a multi-objective optimization pipeline using a tailored hierarchical genetic algorithm (Fig. 1c,d). This approach was chosen because the discrete nature of C-network assignments and the topological constraints of the network make traditional gradient-based optimization methods unsuitable. Our genetic algorithm incorporates custom operators that respect both symmetry and connectivity constraints while exploring the design space. The pipeline simultaneously optimizes three parameters: the assignment of actuators to C-networks (determining which actuators work together), the preset contraction levels of individual actuators (defining actuators' motion), and the temporal actuation sequences (controlling when each C-network activates). This multi-level optimization allows us to find designs that balance the competing demands of control simplicity and task performance while maintaining physical feasibility.

In the seventh paragraph, we expanded the overview of our results to demonstrate not only

the existence of this threshold in C-network performance contribution, but also the system's multi-objective capability and its effectiveness across diverse tasks and topologies. We validate the physical feasibility of our approach through prototyping.

We validate our hypothesis and approach through multiple complementary studies. First, using a complex quadruped metatruss robot (Fig. 1d) tasked with four distinct functions (Fig. 1i, Supplementary Video 1), we demonstrate that a limited number of C-networks can yield competitive performance (Fig. 1j). Our proposed design method achieves a higher ratio of actuatable beams to control units compared to previous VGT systems (Fig. 1k), reducing control complexity while maintaining system capabilities. To demonstrate the diversity of our approach, we successfully applied it to five different truss topologies with various tasks. Finally, we built a physical prototype of one metatruss to validate the physical feasibility and compare its trajectory with the simulation, demonstrating the high accuracy of our simulator (Fig. 1l,m).

Comment #5

The flow of the paper is hard to follow, with methods, computational experiments and physical implementation all blended together. It would be better to reorganise to have the methods together (including all the example problems and their description), then numerical results (optimiser efficacy, convergence, support for synergy hypothesis), then experimental results (with a description of the physical implementation of the device).

Our Response: We have restructured the paper to improve clarity and flow. The manuscript now follows a standard structure with *Introduction*, *Results*, *Methods*, *Discussion*, *Conclusion* sections and *Supplementary Information*. We list the main changes as follows and please refer to the paper for the detailed changes.

In the *Results* section, we added a new Overview subsection that presents the core contributions. The first paragraph introduces the fundamental mechanisms of metatruss. The second paragraph describes the established physical system from our previous work and identifies the key challenges that motivated this study. The third paragraph presents our main contributions, beginning with the simulator development. The fourth paragraph introduces our optimization framework, including the representation, geometric constraints, computational pipeline, and specialized operators. The subsequent paragraphs present our experimental findings, demonstrating the relationship between performance and C-network numbers, the diversity of tasks and topologies achieved, and physical validation of the system.

Throughout the manuscript, we reorganized content to improve readability while preserving key findings. Each section now begins with essential context and motivation before presenting the results. Technical implementations and analyses have been moved to the Supplementary section to help readers focus on the scientific narrative before delving into technical specifics.

We have also separated the original Mechanism and Model section. The *Results-Overview* section now introduces the core mechanism concepts, while the *Methods-Mechanism and Fabrication Details* section presents the technical details. A new Figure 9 illustrates the key system components through diagrams of actuator and joint anatomy, the blocker structure, and the pillbug and quadruped designs. For the simulator content, we created a Results-MetatruSS Simulator section that places our development in context of existing truss robot simulators, with full technical details moved to Supplementary-Simulator Details.

We created a new *Results-Optimization Framework with Tailored Genetic Algorithm* section that introduces the content of design representation, constraints, computation pipeline and tailored operators. We start by introducing existing algorithms for optimizing truss robots, including works optimizing variable geometry truss in the first paragraph. The second paragraph discusses using continuous compact representation such as CPPN, while the third paragraph covers L-system approaches.

We then introduce the specific constraints in our system and explain why we chose to use a genetic algorithm with tailored operators. The mathematical details and implementation specifics of each component are now in dedicated *Methods* subsections: Representation, Constraint, Computation Pipeline and Operators.

In the Design Representation subsection, we present brief definitions and results from the previous Design Representation section. The detailed implementations are now in the *Supplementary-Representation Details* section, which is referenced from the *Results-Design Representation section*.

The C-network Topology Constraints subsection keeps the key definitions of symmetry and connectivity constraints from the previous Tailored Design Configurations and Constraints section. We moved the detailed mathematical definitions to the *Methods-Symmetry Definitions* and *Methods-Constraint Details* sections.

For the computational framework, we created the Multi-objective Computation Pipeline and Tailored Operators subsections from the original Computation Pipeline section. The Pipeline subsection introduces our rationale for NSGA-II and explains the Elite Pool Strategy, which prevents pool domination by elite designs and maintains diversity. The technical details of NSGA-II,

including pareto front and crowding distance definitions, are now in the *Supplementary-NSGA-II Explanation* section.

The Tailored Operators section presents the motivation and implementation of initialization, mutation, and crossover operators, focusing on their management of symmetry and connectivity constraints while preserving exploration capability. The complete implementation details are provided in *Methods-Operators*.

The *Results-Performance with Varying C-network Channel Numbers* section retains its original structure, presenting motivations and assumptions, followed by our approach and detailed results. We moved the technical implementation details and analyses to the *Methods-Numerical Results and Implementation Details* section.

The previous Metatruss Designs and Experimental Validation section has been reorganized into a new *Results-Diversity in Task and Truss Topology* section. The opening paragraph reviews existing work in truss robot tasks and topologies, highlighting their constraints in types of tasks and complexity of topology. We then describe our approach to creating multifunctional structures capable of diverse tasks and topologies, followed by results from four demonstration cases. The *Methods-Truss Topologies and Tasks* section contains the technical definitions of these topologies and tasks, along with their mathematical representation as objective functions.

The new *Results-Physical Validation* section presents our work on the physical 'pillbug' metatruss prototype, previously described in the Evaluation of Pillbug Physical Prototype and Simulation section. Here we present our design and optimization of the physical system, along with comparisons between its physical and simulated walking performance. The *Methods-Physical Prototype and Simulator Accuracy Validation* section contains the complete technical details and analysis.

In the Discussion and Conclusion section, we added three new discussions. The second paragraph presents mechanical logic circuits as a potential future direction for control system simplification. The third paragraph explores extending our approach to non-pneumatic actuation systems, which could potentially remove connectivity constraints. The fourth paragraph examines potential applications of our computational framework to other robotic metamaterials, including 4D printed lattice structures and discrete architectures.

We added a new *Supplementary-NSGA-II Explanation* section detailing our NSGA-II implementation, including the definitions of rank and crowding distances.

In the *Supplementary* section, we added content on On-body Control Circuit Using Mechanical

Logic Gates, which introduces mechanical logic circuits for on-body controllers. This system reduces external components to a single connection by eliminating the need for external control circuits, valves, and additional tubing or wires.

Comment #6

The numerical results lack key information about the optimisation problem and performance. E.g what is the generation size and number of retained elites, how is the diversity metric implemented? What was the computation time?

Our response: Thank you for pointing out the lack of key information. We agree it would be more clear with these numbers included. We added a section Numerical Results and Implementation Details in Methods and cited it the Overview section. This section reports the generation size and number of retained elites and the average computation time. The diversity metric is not an explicit objective. It is achieved through the selection operator implemented based on NSGA-II. It prioritizes the diversity of the generation by computing a crowding distance which tends to keep designs in sparse regions where there are not many designs having the similar performance. In other words, these designs are more unique and contribute to diversity. The details of NSGA-II can be found in Comment #14.

Our modification: We added the following section under Methods and referred to it in *Results-Overview*.

Implementation Details

In our study, we use the notation \leftrightarrow to indicate inter-symmetric channels and \odot to denote self-symmetric channels. We implemented various metatruss designs with different C-network configurations. The quadruped design with 8 channels features a symmetric topology, with channels configured as $0 \leftrightarrow 1, 2 \leftrightarrow 3, 4 \leftrightarrow 5, \odot 6, \odot 7$, comprising 6 inter-symmetric and 2 self-symmetric channels. A simpler quadruped design with 2 channels maintains symmetry with both channels being self-symmetric ($\odot 0, \odot 1$).

We also explored more complex quadruped designs: a 16-channel version with 4 self-symmetric and 12 inter-symmetric channels, a 32-channel version with 8 self-symmetric and 24 inter-symmetric channels, and a 64-channel version with 16 self-symmetric and 48 inter-symmetric channels. The helmet design exhibits symmetry with three self-symmetric channels ($\odot 0, \odot 1, \odot 2$), while the lobster design is symmetric with two inter-symmetric and two self-symmetric channels ($0 \leftrightarrow 1, \odot 2, \odot 3$). Notably, the tentacle design is asymmetric, consisting of four

self-symmetric channels ($\odot 0$, $\odot 1$, $\odot 2$, $\odot 3$).

In our optimization process, we set the active pool size equal to the elite pool size, with an elite percentage of 20% in every generation. For each iteration, we retained 50% designs, generating 35% new designs through mutation and 10% through crossover operations, while introducing 5% new random initializations. The pool size is scaled according to the C-network numbers: 128 for the 8-channel quadruped, 48 for the helmet, 64 for both the lobster and tentacle, 32 for the 2-channel quadruped, 256 for the 16-channel quadruped, 512 for the 32-channel quadruped, and 1024 for the 64-channel quadruped.

We conducted our computations using Google Cloud Computing with 224 cores, achieving an average optimization time of 8 hours and 2 minutes for the 8-channel quadruped over 1000 iterations.

Comment #7

I found the concept and implementation of eliteness somewhat confusing, could you clarify how the ‘elite’ candidates are stored and released.

Our response: In our paper, the elite pool has the same size as the active pool with size N_a . N_a scales with the number of C-networks. The details can be found in the newly added section Numerical Results and Implementation Details. Every generation, all the designs in the active pool are applied with mutation and crossover operators and sorted with NSGA-II, the top $\rho = 20\%$ designs are kept as elites while the rest are discarded. Every $N_g = 5$ generations, the elites are stored in the elite pool and the active pool is initialized with a new generation of designs. Once the elite pool is full, which happens every $N_g/\rho = 25$ generations, all the elites in the pool are moved back to the active pool and continue the optimization.

Our modification: We added a detailed explanation of the elite mechanism in a subsection under Methods titled Elite Pool Strategy for Optimization Across Generations. It is also referred to in the Multi-objective Computation Pipeline subsection under Result. We also clarified in the second sentence that the size of the elite pool is chosen to be the same as the active pool in our paper empirically. Further study will be needed to determine the optimal size of the elite pool. Please see Comment #14 for details.

Comment #8

The implementation of the EA problem is obviously useful, but it’s a sort of brute-force approach. It would be interesting to compare the direct encoding of the networks you’ve used to a shape

grammar or CPPN, which could more compactly encode the design domains and directly embed constraints (rather than randomly searching until a valid solution is found at each iteration). This might allow for larger metatrusses to be generated, or for the morphology (i.e truss configuration) and control to be cooptimized.

Our response:

We appreciate the reviewer’s suggestion about exploring more compact design representations. While the genetic algorithm provides robust performance for complex multi-objective optimization, we investigated neural network approaches to enable continuous optimization in a learned latent space. For the simpler case of fixed-topology optimization, we developed a Graph Attention Network-based Variational Autoencoder (GAT-VAE). Graph neural networks are particularly suitable for this task as they naturally process graph structures and can capture both local edge relationships and global structural patterns - essential features for maintaining C-network connectivity constraints. Given an input truss topology, our GAT-VAE learns to optimize C-network assignments while enforcing connectivity constraints and maximizing shape transformation capabilities.

Rebuttal figure 1: **GAT-VAE Pipeline** begins with an input truss topology and an initial input C-network assign. The GAT-VAE model generates the optimized C-network indices. The optimized C-network follows the connectivity constraint, and its morphed shape has a maximized aspect-ratio.

For this investigation, we focus on the optimization of C-network assignments within 2D truss topologies. Each truss robot with three C-networks has eight possible morphed configurations corresponding to different actuation states (Rebuttal Fig. 2a). We define the aspect ratio of each configuration as the height-to-width ratio of its bounding box, using the maximum aspect ratio across all eight states as a metric for shape-morphing capability. A crucial constraint in our system is C-network connectivity: a C-network assignment is considered valid only if edges sharing the same C-network index form a connected graph component (Rebuttal Fig. 2b). Our dataset comprises 10,000 distinct topologies containing between 1 to 8 triangles, with multiple C-network assignment variations for each topology. The final dataset contains approximately

76% disconnected and 24% connected configurations, where each data point includes initial vertex positions, C-network indices for each edge, connectivity information, and the maximum aspect ratio of transformed shapes.

Rebuttal figure 2: **Aspect ratio and connectivity.** **a.** Each truss robot with three C-networks has eight C-network actuation states, thus having eight transformed shapes based on the state. The height over width ratio of the bounding box of the morphed shape is called the aspect ratio. **b.** A C-network assignment is defined as disconnected, if any C-network is not a connected graph.

The GAT-VAE architecture consists of three main modules (Rebuttal Fig. 3). The encoding module transforms input truss designs into latent vectors through graph attention updates. Each update iteration processes information at three levels: vertex level aggregating C-network information from incident edges, edge level combining information from connected vertices, and global level integrating all vertex and edge features. The decoding module mirrors this structure, taking a truss topology without C-network information and a sampled latent vector to reconstruct the C-network assignments. The property prediction module includes two networks that take the latent vector as input to predict connectivity and maximum aspect ratio. The complete pipeline (Rebuttal Fig. 1) processes each input through four GAT update iterations to ensure sufficient information propagation across the graph structure, with three attention layers handling local, edge-level, and global graph information. Through this architecture, the model learns to capture both the discrete topological constraints of C-network connectivity and the continuous geometric properties affecting shape transformation.

The GAT-VAE achieved good performance across key metrics, demonstrating 99.925% reconstruction accuracy on C-network assignments and 99.999% accuracy in connectivity prediction. The aspect ratio prediction achieved a mean square error of 0.036 relative to a dataset mean of 0.89, indicating precise capture of geometric transformation properties. Analysis of the learned latent space through Principal Component Analysis reveals a clear organizational structure (Rebuttal Fig. 4). The predicted connectivity values show a smooth transition primarily along the first principal component (Rebuttal Fig. 4a), closely matching the actual connectivity labels (Rebuttal Fig. 4b). Visualization of individual latent dimensions shows that most di-

Rebuttal figure 3: **GAT-VAE Model** is composed of the encoding module, decoding module and property prediction module. The **encoding module** takes in a truss design and converts it to a latent vector through GAT updates. The **decoding module** takes in a truss design without C-network information and a latent vector, and uses the same GAT update process to decode the latent vector. The decoded embeddings are reconstructed into the truss design. The **property prediction module** includes two multi-layer perceptron networks that takes in the latent vector and predicts the connectivity and maximum aspect ratio of the truss design.

mensions actively contribute to connectivity categorization (Rebuttal Fig. 4c). The maximum aspect ratio varies smoothly along the second principal component (Rebuttal Fig. 4d,e), notably orthogonal to the connectivity direction. This separation indicates the model has learned to encode geometric transformation properties independently from connectivity features, creating a well-structured latent space suitable for optimization.

To demonstrate the model’s practical utility, we examine both interpolation and optimization capabilities (Rebuttal Fig. 5). The model successfully performs interpolation between different C-network assignments in two scenarios: between two connected configurations, and from a disconnected to a connected configuration (Rebuttal Fig. 5a). In both cases, the intermediate designs show smooth transitions while maintaining physical validity. For connected-to-connected interpolation, the C-network patterns evolve gradually while preserving connectivity. The disconnected-to-connected transition demonstrates the model’s ability to “repair” invalid configurations through the learned latent space navigation. Additionally, we tested the model’s optimization capability by maximizing the aspect ratio of transformed shapes (Rebuttal Fig. 5b). Starting from random C-network assignments, the optimization process leverages gradient information from the aspect ratio predictor to navigate the latent space, producing designs with progressively larger transformation ranges while maintaining connectivity constraints. This gradient-based optimization in the latent space offers significant computational efficiency compared to methods requiring full simulation for each candidate solution.

Rebuttal figure 4: **GAT-VAE latent space visualization.** The latent vectors are projected to two principal dimensions using PCA. **a.** The predicted graph connectivity value shows a smooth transition mainly along the first principal component dimension. **b.** The actual label of graph connectivity shows that the model clearly and correctly separates the latent vectors into connected and disconnected categories. **c.** 80 out of 128 latent dimensions are visualized individually, showing most of the dimensions are learning meaningful information about the connectivity. **d.** The predicted maximum aspect ratio shows a smooth transition mainly along the second principal component dimension, perpendicular to the direction of connectivity prediction. **e.** The actual maximum aspect ratio is very close to the predicted maximum aspect ratio with small difference.

While the GAT-VAE demonstrates promising results for fixed-topology optimization, our genetic algorithm approach remains more suitable for the full scope of our design problems. The neural network approach offers advantages through its continuous latent space representation and rapid evaluation capability, particularly for exploring local design variations within a fixed topology. However, it is currently limited to fixed topologies and single objectives, whereas our genetic algorithm framework handles variable topologies, multiple objectives, and complex constraints simultaneously. The genetic algorithm's ability to directly encode these features, combined with our tailored operators for maintaining connectivity and symmetry constraints, provides more robust performance for the complex multi-objective tasks demonstrated in our paper.

It would be interesting to compare the direct encoding of the networks you've used to a shape grammar or CPPN

CPPNs have demonstrated significant success in evolving complex morphologies, particularly in systems with regular spatial structures. They have been effectively used for evolving soft voxel-

Rebuttal figure 5: **GAT-VAE interpolation and optimizations.** **a.** The model is able to interpolate the C-network assignments between two trusses with the same connectivity (interpolation 1) or different connectivity (interpolation 2). **b.** The model optimizes the C-network assignment towards a larger maximum aspect ratio, which shows a continuous improvement on the aspect ratio of the transformed shape.

based robots [6], generating 3D printable objects [35], and creating complex patterns [36]. However, our truss optimization problem presents fundamentally different challenges that make CPPNs less suitable for our application.

Our analysis identified three key limitations of CPPNs for truss optimization. First, unlike the regular voxel grids used in [6], our truss system represents an irregular graph structure where topological relationships between edges do not map naturally to coordinate-based representations. Second, while CPPNs excel at generating continuous patterns with natural variation [36], our system requires discrete C-network assignments with strict connectivity constraints. Converting continuous CPPN outputs to discrete assignments while maintaining valid connected components would introduce significant computational overhead and potential instability. Third, CPPNs are designed to capture global patterns and regularities [35], but our optimization requires specific local structural variations and edge-to-edge relationships that are fundamentally topological rather than spatial.

These limitations represent fundamental mismatches between CPPN’s strengths and our problem requirements rather than technical implementation challenges.

Comment #9

Another interesting feature is the idea of muscle synergy you highlighted: your system clusters together groups of ‘muscles’ to reduce the total number, but each has an independent control signal. Can you extend this to show how the muscles can coordinate by responding a central signal? i.e have a single repeating signal that coordinates all the muscles in the way that a regular heart beat triggers a series of muscles to expand and contract in sequence.

Our response: We sincerely appreciate your insightful suggestion about coordinating muscles with a central signal. This concept aligns perfectly with biological muscle coordination, where complex sequential motions like walking often emerge from a single central pattern generator that coordinates multiple muscle groups through periodic signals.

Drawing inspiration from this biological paradigm, we demonstrate how our optimized C-networks can be coordinated through a single mechanical “central pattern generator” implemented using pneumatic logic circuits. Recent work with pneumatic-powered walking robots controlled by mechanical circuits demonstrates the feasibility of such untethered pneumatic robots, though these circuits were manually designed and limited to a single motion.

Taking our pillbug’s forward walking motion as an example, our optimization generated a 4-bit binary control sequence over 4 time steps (Figure 13a). Theoretically, this sequence can be implemented using a 2-to-4 multiplexer (MUX) where a 2-digit input (representing time step) generates a 4-digit output (representing C-network states). Using Karnaugh maps for logic optimization, we derived a simplified circuit requiring only 14 logic gates including one clock unit (Figure 13c).

The physical implementation leverages pneumatic mechanical logic gates that use pneumatic energy as both power and signal [61, 62]. Each logic unit consists of a cylindrical chamber with a bistable membrane (Figure 13d), where two chambers are separated by the membrane with three ports on each side. Two ports are connected through a rubber air tubing going through the chamber, while one port directly connects to a chamber. The binary membrane can pop up or down (Figure 13e), blocking the air channel in the corresponding chamber. When the pressure difference between chambers exceeds a threshold, the membrane rapidly switches states.

Through different port configurations for constant air source and signal air source, these units can function as AND, OR, and NOT gates (Figure 13f). Most importantly, by connecting an external air tank as a reservoir (Figure 13g), the unit becomes an oscillator generating periodic signals - analogous to a biological central pattern generator. The oscillation period can be tuned by adjusting air flow rate, pressure, and chamber volume.

Rebuttal figure 6: (Figure 13 in manuscript) **On-body control circuit using mechanical logic gates.** **a**, Open-loop control signal optimized for Pillbug metatruss towards walking forward. **b**, 4-to-8 multiplexer circuit for the open-loop signal. **c**, Simplified circuit for the open-loop signal. **d**, The basic mechanical circuit unit. **e**, The illustration of the bi-stable mechanism of the mechanical circuit unit. **f**, The illustration of the ports connection for AND, OR, and NOT gate. **g**, The illustration of the clock unit. **h**, A rendering of the pillbug with on-body mechanical open-loop control circuit.

While current units measure approximately 3.25cm in diameter, miniaturization appears feasible through reduction of membrane, tubing, and chamber sizes, with corresponding adjustments to pressure, flow rate, and membrane properties. The metatruss design is particularly suitable for incorporating such control circuits due to its inherent scalability and volumetric internal space. Assuming the logic units could be miniaturized to 1cm^3 and metatruss beams scaled to 20cm

length, we created renderings demonstrating how a pillbug robot could carry its own control circuit board (Figure 13h). In this configuration, the robot would require only a single constant air pressure source - either through a tethered tube or potentially untethered with an onboard compressed air tank.

This mechanical implementation of centralized control demonstrates how our metatruss approach can move beyond simple actuator grouping toward truly coordinated, self-contained robotic systems. Just as biological muscle synergies reduce neural control complexity through coordinated activation patterns, our mechanical logic approach reduces control complexity through physical implementation of optimized actuation sequences.

We're excited about the potential of this approach to significantly simplify control mechanisms and enable more autonomous, untethered soft robots, though we acknowledge that future work would be needed to fully implement and validate this idea. We're grateful for your suggestion, as it has prompted us to explore this promising direction further.

Comment #10

The figures were often used as evidence for features which were not shown or hard to understand – they need to be modified to give a clearer explanation of the optimisation framework. E.g.:a. “These joints have inner air channels (Fig. 1a)” – 1a illustrates the idea of shape change using VGTs, but no air channels are shown

Our response: Thanks for pointing that out. We agree with the confusion here, and we modified the figure 1 and added more details of the metatruss features from the previous work in Mechanism and Fabrication Details under Methods section.

Our modification: We have revised Figure 1 to clearly show both the mechanism and optimization framework. In Figure 1a, we show a double tetrahedron metatruss with semi-transparent exterior material to reveal the inner air channels. Each channel is color-coded by C-network, and we use solid circles to show closed inflation valves (negative pressure, semi-transparent channels) and hollow circles to show open inflation valves (high pressure, opaque channels), directly indicating when beams expand or contract. Figure 1b demonstrates the eight possible morphological configurations achievable through different combinations of these valve states, now rendered with consistent joint visualization to match 1a.

Figure 1c illustrates our tailored genetic algorithm framework, showing how NSGA-II enables multi-objective optimization through the Pareto front approach. The left side shows the input topology, which is given and fixed, while the right side shows one optimized design from the

Rebuttal figure 7: Figure 1 Modification.

Pareto front that is expanded in subsequent figures to show our three key optimization parameters:

- **C-network Assignment:** Figure 1d shows how actuators are grouped into different C-networks. Figure 1e shows a detailed visualization of the selective air channel joint structure. Figure 1f provides a mechanical rendering of the blocker structure that enables discrete contraction ratios.
- **Contraction Parameters** are shown in Figure 1g, displaying the discrete contraction ratios (0.0, 0.12, 0.24, 0.36) assigned to each actuator
- **Control Parameters** are illustrated in Figure 1h through the binary control signals defining when each C-network activates

We have expanded the Mechanism and Fabrication Details section in Methods to provide comprehensive information about these mechanical features from our previous work.

Comment #11

“metatruss system also incorporates the design of a discrete-preset contraction beam (Fig. 1c).”
 – Fig 1c doesn’t really show anything in my opinion, illustrating this preset configuration needs more than a bar chart with discrete steps. For both this and 6a it would be beneficial to show the mechanical implementation of the device rather than a very loose representation (and better yet move that to an experimental implementation section)

Our response: We agree it would be beneficial to incorporate the mechanical implementation details of the metatruss system. We added figure 1f which provides a clear illustration of the discrete-preset contraction mechanism, showing how inserting a blocker structure at three possible positions on the piston component determines the maximum contraction ratio of the actuator when negative pressure is applied. Additionally, we added Methods - Mechanism and Fabrication Details section that documents the complete metatruss system. This new section details the design and fabrication of all core components: pneumatic actuators based on modified syringes with bidirectional air ports, 3D-printed selective-air-channel joints with optimized inner air channel geometry, C-shaped blockers, end caps, and the complete assembly process using super glue and friction-fit rubber tubing connections. Each joint's inner air channels are generated based on C-network assignments and optimized using Kangaroo for minimum separation between channels while maintaining air-tightness and facilitating post-processing. While the underlying mechanisms build upon those introduced in PneuMesh, we present complete implementation details here for clarity and reproducibility.

Our modification: In figure 1f, we illustrated the discrete-preset contraction mechanism through a cross-sectional view showing the blocker insertion process and its effect on maximum contraction ratio. The new *Methods-Mechanism and Fabrication Details* section includes rendered figures showing: (1) component-level designs including actuator parts, joint structure with optimized air channels, and assembly elements, (2) step-by-step assembly procedures from individual components to working actuators to complete joints, and (3) fully assembled metatruss robots including both the quadruped and pillbug configurations.

The Actuator Design subsection now specifically details the implementation of the discrete-preset beam system using 3D-printed pistons with positioning holes and blockers, complementing the overview provided in figure 1f.

Comment #12

The writing seems to conflate the actuator layout with truss design, it should be clear that the truss itself (what I would call the morphology) is not optimised, just the actuator layout and cycle (the controller, in my opinion). E.g Fig 4a shows topology as part of the optimisation, presumably this means C-network topology, it needs to be clarified that this is not VGT topology

Our response: Thanks for pointing out the confusion about the terminology “topology”. You are right, the truss topology is a fixed input parameter, and the C-network symmetry (C-network topology constraints) is another input parameter. The optimizer's goal is to determine the actuator layout through finding optimal C-network indices assigned to each beam, discrete contraction ratios, and binary control signals for each C-network across time.

Rebuttal figure 8: (Figure 9 in manuscript) Metatruss Mechanism

Our modification: We clarified this distinction in two places in the manuscript:

In the Overview section under Results, we explicitly state the input and output parameters: “Given a metatruss topology, initial joint positions, tasks, number of C-networks, and C-network symmetry, our optimizer finds the optimal C-network assignment, contraction levels, and actuation signals to maximize the metatruss’s multi-objective performance across the specified tasks.”

Rebuttal figure 9: Figure 4 modification 1.

In Figure 4a, we updated the diagram to clearly distinguish input parameters from optimization variables by (1) adding a labeled “Input Setting” box containing the metatruss topology (edge connections between vertices), initial vertex positions, symmetry constraints, C-network configurations, and objectives, and (2) separating these fixed inputs from the parameters being optimized through the genetic algorithm

Comment #13

Figure 4(vii) gives the impression that fitness is reevaluated after the mutation/crossover step, but (if I’m not mistaken) this occurs at 4(iv)

Our response:

We modified Figure 4 in several ways. First, we removed step vii and created a direct connection from step vi back to the active pool, illustrating that mutation, crossover, and initialization operators directly generate the next generation of designs. Second, we moved Figure 4b to the bottom left corner. Third, we moved the previous step vii to become Figure 4c in the bottom right corner. This new Figure 4c illustrates the multi-objective performance of designs in the current generation, where some designs generated through genetic operators demonstrate performance superior to the current Pareto Front, showing how the controlled randomness in these operators can lead to improved solutions.

Comment #14

“sorting them using NSGA-II based on their rank and crowding distance.” – crowding distance is not defined here but is key to the idea of eliteness

Rebuttal figure 10: Figure 4 modification 2.

Our Response: We agree that defining the crowding distance is crucial for understanding the concept of eliteness in NSGA-II. We have added a detailed explanation of both the non-dominated sorting and crowding distance calculation in *Supplementary Note 4: NSGA-II Explanation*, including a new figure to illustrate these concepts.

The crowding distance measures the density of designs surrounding a particular design in the fitness space. In our revised manuscript, we explain that NSGA-II first calculates the non-domination rank (R) for each design, and then calculates the crowding distances (CD) within designs of the same rank.

We've added the definition and formula for the crowding distance, showing it as the sum of normalized distances between a design and its nearest neighbors in each fitness dimension. This selection approach identifies designs that maintain diversity in the Pareto front.

We've also included an explanation of the hypervolume metric for evaluating the overall performance of a generation in multi-objective optimization.

We referenced this section in the Multi-objective Computation Pipeline section under Results.

Finally, we added figure 12 to show the steps of the NSGA-II algorithm, which first sorts designs into non-dominated fronts, then calculates crowding distances within each front, and finally selects designs based on both rank and crowding distance.

Our modification: We added *Supplementary Note 4: NSGA-II Explanation*, which explains the motivation, definition and calculation method of non-dominance rank, crowding distance and hypervolume.

NSGA-II introduces metrics to find the best-performing group on multi-objectives. It uses non-dominant sorting and crowding distance sorting. First, it calculates the rank (R) of a design based on whether it is dominated by other designs. The definition of dominance is as follows: Given a design 1 with multi-objective fitness $S_1 = s_{1,0}, s_{1,1}, \dots, s_{1,n}$, and a design 2 with fitness $S_2 = s_{2,0}, s_{2,1}, \dots, s_{2,n}$, if

$$s_{2,i} \geq s_{1,i} \forall i \in 0, 1, \dots, n \text{ and } \exists j : s_{2,j} > s_{1,j}$$

we say that design 2 dominates design 1. Designs that are not dominated by any other design have rank 0. To determine rank 1, we consider only the designs not in rank 0, and find those that are not dominated within this subset. We continue this process to find subsequent ranks (Fig. 12a). The $R = 0$ designs form the Pareto set, and their fitness values form the Pareto front.

The crowding distance (CD) is calculated to further sort designs within the same rank and to encourage diversity. CD measures how crowded it is around a design in the fitness space. It is calculated as follows: For a design i in a particular front:

For each fitness dimension m : a. Sort the designs in the front by fitness m . b. Assign infinite distance to boundary designs. c. For all other designs, assign a distance equal to the absolute normalized difference in the fitness values of two adjacent designs. The overall CD for design i is the sum of individual distance values for each fitness dimension:

$$CD_i = \sum_{m=1}^M \frac{f_m(i+1) - f_m(i-1)}{f_m^{max} - f_m^{min}}$$

where M is the number of fitness dimensions, $f_m(i)$ is the m -th fitness value of the i -th design, and f_m^{max} and f_m^{min} are the maximum and minimum values of the m -th fitness dimension. A larger CD implies that the design is more unique and potentially has distinct features, adding to the diversity of the generation. During the selection process, designs are first sorted by rank and then by crowding distance (Fig. 12b).

Rebuttal figure 11: (Figure 12 in manuscript) **NSGA-II explanation for a two-fitness optimization problem.** **a**, Non-dominated sorting: Designs 1 and 2 have $R = 0$ as they are not dominated by any other design. Design 3 is dominated by Design 1 but not by any other non- $R = 0$ design, so it has $R = 1$. **b**, Crowding distance calculation: For each design, the cuboid formed by its nearest neighbors in the fitness space is considered. The crowding distance is the sum of the normalized side lengths of this cuboid. For example, $CD_1 = \frac{d_{1,2}}{f_2^{max} - f_2^{min}} + \frac{d_{1,1}}{f_1^{max} - f_1^{min}}$, where $d_{1,2}$ and $d_{1,1}$ are the distances to the nearest neighbors in fitness dimensions 2 and 1, respectively. **c**, Hypervolume: The hypervolume (area in 2D) covered by the Pareto front, which serves as a measure of the quality and diversity of non-dominated designs.

Reviewer #2 (Remarks to the Author):

Comment #15

I congratulate the authors to this delightful paper and highly interesting approach and novel methodology.

Overall paper: Very good paper excellent approach.

Language: High level.

Abstract: - Precise abstract with all important points

Our response: Thank you for the kind words!

Comment #16

Biological Inspiration and Our Hypothesis: - Page 3 figure 1g please increase the font size of the references to match the size of the font in 1f. In addition please standardize the axis titles as there is a mix of words starting with capital letters (c,f, etc.) and in (d) only the first word starts with a capital letter - Page 4 line 79 there is an “an” missing before “empirical”.

Our response: Thanks for pointing out the small font sizes and the grammatical mistake.

Our modification: We changed citations in figure 1k from abbreviations to citation numbers and

changed the font size to 7 to match other subfigures.

Rebuttal figure 12: Figure 1 modification 2.

Comment #17

Mechanism and model: - Page 4 line 88 change: “validation for” to “validation of” - Please add an explanation of “NSGA-II”.

Our response: Thanks for pointing out the grammatical mistake and we agree that an explanation of NSGA-II would be helpful.

Our modification: We revised the writing of the sentences in the first paragraph of Mechanism and Fabrication Details for clarity and conciseness, and removed the phrase “validation for”.

We added a NSGA-II Explanation section under Methods, which explains the motivation, definition and calculation method of non-dominance rank, crowding distance and hypervolume. Please refer to Comment #14 for details.

Comment #18

Performance with Varying C-network Channel Numbers: - Figure 6 switch the lettering of subfigures c) and d)

Our response: Thanks for suggesting a clear reordering of the figure labels. We’ve switched the lettering of c and d.

Our modification: We modified the label and the captions of figure 6 c and d.

Rebuttal figure 13: Figure 6 modification.

Comment #19

Metatruss Designs and Experimental Validation: - If possible I would like to see a physical demonstrator for the shape shifting helmet. - Page 16 line 388 please change Figure 7g-h to 7g,h as used above for 7d,e (page 15 line 378). - Page 17 line 415 there is a space missing Fig.8a - The pillbug demonstrator is very impressive. Although it was not the focus of the experiment but could you please measure the walking speed of the demonstrator and add it to the text?

Our response: Thank you for your interest in the physical demonstrator of the helmet! We elected not to create a physical prototype of the helmet for several reasons. It has more beams and joints than the pillbug, and the shape transformation is more straight forward than the pillbug's locomotion due to the deterministic truss topology, which matters more to the simulator performance. With limited time, we chose to implement the pillbug because we want to know whether the dynamic motion of the pillbug matches the simulation or not. We agree given enough time for implementing prototypes, a few more would be ideal and we are excited to continue exploring these designs in future work. We do see the major contribution of this paper more on the optimization and computational design.

We appreciate you pointing out the format issues and suggesting including the pillbug locomotion speed. We have corrected the format and included the measured moving speed in the text.

Our modification: We changed g-h to g-i under *Methods-Lobster Robot Trained for Energy Efficiency* section, and verified the consistent usage of figure label formatting throughout the manuscript.

Figure 7g-i shows the lobster performance with iterations. We observed that, although all the metrics are improving, using multiple objectives can enhance the search efficiency and quality. For example, when searching for a solution for both

the locomotion task and energy efficiency task, the search is faster and converges at a better result than searching for locomotion only (Fig. 7h).

We ensured consistent spacing between 'Fig.' and all subsequent figure subpanel labels throughout the manuscript.

We measured the average moving speed of the pillbug by dividing the total displacement by the total operating time from the three tracking trajectories and updated the values in the Physical Prototype and Simulator Accuracy Validation Section under Methods.

For reference, the fully contracted length of each beam in the prototype is 17.3 cm and the fully extended length is 24.5 cm. The average locomotion speed is 2.45 cm/s. The total displacement is 86.0 cm.

Comment #20

Conclusion: - Please briefly elaborate on the chosen examples and the results achieved with them in terms of expectance, performance and usefulness as system capabilities demonstrations. Could there be other demonstration cases?

Our response: We appreciate your suggestion of adding higher-level descriptions of the expected performance, actual performance, and the usefulness interpretation of the demonstrations, as well as the potential extra demonstration cases. To keep both the high-level descriptions and implementation details as well as keeping the structural clarity of the paper, we added a new paragraph, Diversity in Task and Truss Topology, under Result, which includes the brief summary of the goal, the expectancy, and the results of the demonstrations. We moved the detailed implementation and numerical result into the Truss Topologies and Tasks section under Methods and revised the writing for clarity.

Our modification: We added a new section *Results-Diversity in Task and Truss Topology*:

Diversity in Task and Truss Topology

To demonstrate the versatility of our metatruss method, we explored a variety of truss topologies and functional objectives beyond simple locomotion tasks. Previous work on Variable Geometry Trusses (VGTs) has primarily focused on single-function designs or limited morphological changes due to control complexity [18, 21]. Similarly, other morphing robots have typically been optimized for specific tasks such as locomotion on different terrains or in water [5, 37]. Traditional limbed

robots, while versatile in movement, are limited in their ability to perform significant shape changes [38].

Our method, in contrast, enables the design of multi-functional, highly adaptable structures while maintaining a simplified control system. It allows for both complex locomotion and volumetric shape morphing. This capability sets our approach apart from both traditional VGTs and limbed robots.

We hypothesized that our approach could optimize trusses for diverse, potentially conflicting objectives within a single design, including both locomotion and shape-approximation tasks. To test this, we developed four distinct examples: a quadruped robot, a shape-shifting helmet, a lobster-inspired walking robot, and a tentacle-like actuator (Fig. 1b, Fig. 7).

The quadruped robot was optimized for four motion objectives: walking, turning, tilting, and crouching (Fig. 1e, Supplementary Video 1). This demonstration served two purposes: first, to show that the metatruss could achieve traditional robotic tasks like locomotion and pose changes, and second, to demonstrate that our computational pipeline enabled multiple tasks with a single physical configuration. The robot successfully performed all four motions, with performance improving as the number of C-networks increased up to a threshold value (Fig. 6b).

The shape-shifting helmet (Fig. 7a-d) demonstrated our method's capability for precise volumetric shape morphing, successfully transforming between two distinct target shapes while maintaining structural integrity. This capability, which is not typically achievable with traditional limbed robots, could enable new robotic functionalities, from adapting morphology to meet different environmental constraints and functional requirements to precisely approximating different shapes for aesthetic purposes.

The lobster-inspired robot (Fig. 7f-i) incorporated energy efficiency alongside locomotion speed, demonstrating improved walking performance and optimization efficiency compared to single-objective optimization. This multi-objective approach extends beyond typical terrain-specific optimizations that focus solely on speed, demonstrating the potential for sustainable locomotion in robots with numerous actuators.

The tentacle-like actuator (Fig. 7j-l) achieved high precision in reaching multiple 3D target positions, with error rates below 1e-2mm for a 173mm beam length,

demonstrating the method's capability for precise shape control. This accuracy suggests promising applications in high-precision manipulation tasks using truss robots.

These examples demonstrate our method's ability to optimize complex, multi-functional truss designs capable of both locomotion and significant shape changes while maintaining simplified control. Our results consistently met or exceeded performance expectations, highlighting our approach's effectiveness in creating versatile, adaptive robotic systems that bridge the gap between traditional limbed robots and highly deformable structures. For detailed information about the topologies and tasks, refer to Truss Topologies and Tasks.

Pillbug Robot

While the previous examples demonstrate the diversity of achievable morphologies and tasks, the Pillbug robot was designed specifically to showcase the physical feasibility of our metatruss design and to assess the accuracy of our simulation. The Pillbug was optimized for two tasks similar to those of the quadruped robot: walking forward and lowering its body.

The resulting performance of the quadruped, pillbug and the rest demonstrations are recorded in Performance with Varying C-network Channel Numbers, Physical Prototype and Simulator Accuracy Validation and Diversity in Task and Truss Topology sections respectively.

Comment #21

Methods: - Figure 9 fonts and font sizes are all different please standardize. - Page 23 line numbers are missing for the first paragraph and the figure reference is wrong should be Fig. 9 not fig.10 as Figure 10 describes a different system.

Our response: Thank you for pointing out the font mismatch, the missing line numbers and the reference errors. We've corrected them and updated the texts.

Our modifications: We synced the font style and font size in the figure.

We fixed the missing line numbers and made sure that it works for the entire manuscript.

We fixed the figure reference numbers to the original figure 9 and 10, and made sure the references are correct through the manuscript.

Rebuttal figure 14: (Figure 11 in manuscript) Simulation explanation figure modified.

Reviewer #3 (Remarks to the Author):

Comment #22

The authors develop a novel framework and optimization approach to design multifunctional truss systems and validate in on an experimental pneumatically driven pill bug. The approach seems like a good advance for this field and seems like a good fit for the journal.

Our response: We appreciate your kind words!

Comment #23

The only feedback I have is for the authors to consider other emerging and related systems where this work seems to be applicable, such as 3D printed robotic metamaterials (<https://www.science.org/doi/10.1126/science.abn0090>)

and 4D printed lattice-based robots (<https://onlinelibrary.wiley.com/doi/abs/10.1002/adma.202307858>), and discuss how these tools could be applied for such systems.

Our response: Thank you for suggesting the two papers. They are highly relevant because both the functional lattice and the truss have a complex discrete network structure, where the property or type of unit can affect the global performance and external motions. With a proper optimization process, these robotic structures can be able to perform more complex tasks or show accurate performances without manual design. We've included the two works in our paper.

Our modification: We added the following paragraph in the Conclusion section to discuss the potential usage of our approach in emerging multi-functional robots with complex discrete structures.

Our method shows potential for application in other emerging fields of robotic metamaterials and structures. Recent work has introduced strategies to design and construct classes of robotic metamaterials and 4D printed lattice structures that

incorporate complex, multifunctional elements in discrete architectures [39, 40]. These approaches create materials capable of outputting multi-DoF motions, sensing capabilities, and programmable thermal and mechanical responses through the manipulation of the properties of local discrete units within 2D or 3D lattices. Our tailored multi-objective genetic algorithm, originally developed for metatruss optimization, could be adapted to optimize these lattice-based structures and potentially automate and speed up the design process, optimizing the arrangement and properties of discrete elements to achieve more complex macro-scale performances or motions while respecting manufacturing and material constraints.

Comment #24

For some reason I could not view or run the code. Instructions on this would be most helpful.

Our response: We appreciate you sharing this code issue. We've fixed the code repository and updated it on github. You can check README.md to run the code. Please don't hesitate to email us if there is any problem running the code.

<http://riceroll.github.io/metatrusscode>

Response to Reviewers

This document contains our point-by-point responses to all reviewer comments and concerns. For clarity of presentation, we use the following color coding scheme:

Reviewer comments are shown in black text. Our responses to these comments are highlighted in blue. The changes made to the manuscript and supplementary materials are indicated as follows: light green text denotes minor revisions and unchanged text, while green text highlights substantial modifications and key changes.

We greatly appreciate all the kind words and valuable feedback provided by the reviewers! We have addressed each comment and quoted the modifications we made.

Referee #1 (Remarks to the Author):

Comment #1

Thank you for the comprehensive feedback, I believe the authors have done a commendable job of addressing my concerns from the first version.

The work was always technically very sound, but in my opinion needed to clarify the methodology and contributions. I believe this has now largely been achieved.

Our Response: We thank the referee for the kind words!

Comment #2

My only outstanding comments are: 1) I think it would be valuable to define the meaning of 'morphology' and 'topology' in this paper early on as this would immediately dispel any confusion about which parts of the network/structure are included in the optimization

Our Response:

Thank you for pointing that out. We agree that the two terms would be confusing without clarification, and it would be beneficial for readers to see clear definitions early in the paper. We have inserted definitions of truss topology and truss morphology, and clarified that while the overall truss topology is given as input, we optimize the C-network topology (assignment), such that the resulting sequential morphological changes can achieve the desired objective behaviors.

Our Modifications:

We inserted the following highlighted explanation after the third paragraph in Methods-Overview.

A metatruss can achieve specific shapes or perform sequential motions through activation signals of its C-networks, enabling tasks that require locomotion or shape changes. Whereas previous work [1] demonstrated hand-designed C-network assignments and actuation signals for given tasks, our work automates this process for more complex truss topologies and diverse tasks. Given a metatruss topology, initial joint positions, tasks, number of C-networks, and C-network symmetry, our optimizer finds the optimal C-network assignment, contraction levels, and actuation signals to maximize the metatruss's multi-objective performance across the specified tasks. For a detailed problem definition, refer to ???. The specific truss topologies and tasks explored in this paper are described in ???.

For clarity, in this paper, metatruss topology refers to the connectivity and structural relationship between joints and beams, analogous to a graph structure. The topology represents the fundamental structure that remains fixed after design, including which beams connect to which joints and how they're grouped into C-networks. Morphology refers to the physical shape and form that the structure takes when the beams change length. As actuators in the metatruss expand or contract, the positions of joints shift while maintaining their topological connections, creating different morphologies that enable the robot to perform various tasks. In this paper, we have a fixed truss topology as input, and we aim to optimize the topologies of C-networks that comprises the entire metatruss, such that the resulting sequential morphological change can achieve the objective behavior.

To optimize the metatruss design, we developed a highly-damped dynamical simulator that balances computational efficiency with physical accuracy. While existing simulators like Finite Element Methods offer high fidelity but are computationally intensive, and pure kinematic approaches are fast but oversimplified, our approach strikes a middle ground necessary for evolutionary optimization. The simulator approximates quasi-static behavior through significant damping while accurately cap-

turing essential physical interactions including length constraints, gravity, ground collision, and friction. This design choice enables rapid evaluation of thousands of design iterations while maintaining sufficient accuracy for real-world usage, as validated through our physical prototypes. The simulator’s performance and accuracy are thoroughly examined in Simulator, where we demonstrate comparable accuracy to established physics engines while achieving substantially faster computation times necessary for our genetic optimization pipeline.

Comment #3

2) The phrase metatruss seems to be used in several contexts. i.e in the abstract metatruss is referred to as "an optimization framework", but in the remainder of the paper it seems to refer to the VCG topology and control. Could you clarify its intended meaning? Also the term should be defined in the main text

Our Response: Thank you very much for catching this inconsistency. We found the confusing sentence appears in the abstract where we incorrectly referred to metatruss as “an optimization framework”. We’ve clarified this concept with following modification.

Our Modifications:

The original text in the abstract:

A robot’s morphology is pivotal to its functionality, as seen in natural systems where octopi squeeze through small apertures and caterpillars use peristaltic shape changes to navigate diverse environments. While existing robotic systems struggle to achieve precise volumetric transformations, Variable Geometry Truss (VGT) systems show strong morphological capabilities resulting from their large numbers of actuating beams, enabling volumetric shape changes. However, current VGTs face scalability issues due to control complexity that increases exponentially with the number of beams, resulting in physical implementations limited to either a few tetrahedral units or minimal actuatable beams. Previous work [1] simplified control by grouping actuators into synchronized subnetworks through shared air channels, yet relied on manual design for network assignments and control sequences. We introduce metatruss, an optimization framework that builds upon this actuation mechanism through a tailored multi-objective genetic algorithm to automate the computation of actuator grouping, contraction ratios, and control sequences. We developed a highly-damped dynamical simulator that balances computational efficiency with physical accuracy, validated through experimental prototypes. Through diverse robotic implementa-

tions, we show that metatruss can achieve complex morphological adaptations while maintaining control simplicity. Empirical results validate the hypothesis that there exists an optimal number of control networks for a given metatruss, beyond which additional networks yield diminishing returns in performance across multiple tasks.

The modified text in the abstract:

A robot's morphology is pivotal to its functionality, as seen in natural systems where octopi squeeze through small apertures and caterpillars use peristaltic shape changes to navigate diverse environments. While existing robotic systems struggle to achieve precise volumetric transformations, Variable Geometry Truss (VGT) systems show strong morphological capabilities resulting from their large numbers of actuating beams, enabling volumetric shape changes. However, current VGTs face scalability issues due to control complexity that increases exponentially with the number of beams, resulting in physical implementations limited to either a few tetrahedral units or minimal actuatable beams. Previous work [1] **introduced metatruss, a type of truss robot with** simplified control by grouping actuators into synchronized subnetworks through shared air channels, yet relied on manual design for network assignments and control sequences. We introduce **an optimization framework for metatruss** through a tailored multi-objective genetic algorithm to automate the computation of actuator grouping, contraction ratios, and control sequences. We developed a highly-damped dynamical simulator that balances computational efficiency with physical accuracy, validated through experimental prototypes. Through diverse robotic implementations, we show that metatruss can achieve complex morphological adaptations while maintaining control simplicity. Empirical results validate the hypothesis that there exists an optimal number of control networks for a given metatruss, beyond which additional networks yield diminishing returns in performance across multiple tasks.

Comment #4

3) Several of the section links are broken in the overview section (lines 149-165)

Our Response: Thank you for pointing this out. We have fixed all the section links.

Referee #2 (Remarks to the Author):

Comment #5

Dear authors,

Thank you for responding to all comments and changing the manuscript accordingly. The only issue that is left is that in the overview the references seem to be off there is ?? were a number should be. Otherwise all comments were met and the manuscript was improved above and beyond what was requested. Especially the on-body control system is an intriguing idea.

Nothing to add.

Our Response: We really appreciate all the positive comments! Thanks for all the constructive suggestions that helped us improve the clarity and quality of our manuscript.

The section links are fixed. Please refer to comment Comment #4 for details.

Referee #3 (Remarks to the Author):

Comment #6

I appreciate the time and care the authors put in their responses to reviewers.

Our Response: We sincerely appreciate your thoughtful feedback and encouragement!

Comment #7

1. I could not see the authors' need for their rebuttal figure 1 (i.e., I did not find anywhere in their text where they reference rebuttal figure 1.

Our Response: Thank you for your observation. We have removed Note 8 and Rebuttal Figure 1 from the revised manuscript. These materials were initially included to provide a brief overview of our early exploration of GAT-VAE. However, upon further consideration, we believe that a more thorough investigation of this direction would be more appropriate as part of a dedicated future study. To maintain the clarity and focus of the current work, which centers on our genetic algorithm-based optimization approach, we now briefly mention the limitations of GNN-based methods in the revised manuscript, without including the detailed figure or supplementary note.

Our Modification:

We added the following highlighted paragraph in Results - Optimization Framework with Tailored Genetic Algorithm:

Other approaches such as the use of transformers [2, 3] or L-systems [4] for tree-topology robots also face limitations when applied to metatruss designs. These methods are well-suited for acyclic, tree-like structures but struggle with the cyclic topology of trusses. Moreover, the number of edges in our metatruss is significantly larger than in typical limbed robots, adding another layer of complexity to the encoding and optimization process.

Graph Neural Networks (GNNs) are naturally suited for cyclic topologies like those in our metatruss system. However, information degradation during message passing has long been a bottleneck [5], and for truss structures, which are supposed to be scalable, the generalizability to more complex structures remains challenging for GNNs.

Given the unique challenges of metatruss optimization—including C-network connectivity constraints, cyclic graph topology, and multi-objective requirements—existing implicit encoding approaches prove inadequate. Instead, we opt for discrete optimization methods, specifically genetic algorithms, which allow direct optimization on explicit encodings. The flexibility of genetic operators enables us to tailor them to our specific constraints. To address the multi-objective nature of our problem, we implement the NSGA-II algorithm (see ??), facilitating simultaneous optimization of metatruss designs across multiple performance criteria.

Comment #8

2. I noticed that, in their responses to the second reviewer, the authors added locomotion speed. I appreciate this, and think that it might be even better to report the speed in terms of body length per unit time so they can have a direct comparison to biology.

Our Response:

Thank you for this suggestion. We agree that reporting speed in terms of body length per unit time provides a more intuitive comparison.

Our Modification:

We have added the total body length and expressed the locomotion speed as a percentage of

body length per second to facilitate direct comparisons.

For reference, the fully contracted length of each beam in the prototype is 17.3 cm and the fully extended length is 24.5 cm. **The average total body length during movement is 51.4 cm**, with an average locomotion speed of 2.45 cm/s (approximately 4.7% of body length per second). The total displacement is 86.0 cm.